# Fine-mapping of prostate cancer susceptibility loci in a large meta-analysis identifies candidate causal variants

Tokhir Dadaev (iD), Edward J. Saunders et al.[#]

Prostate cancer is a polygenic disease with a large heritable component. A number of common, low-penetrance prostate cancer risk loci have been identified through GWAS. Here we apply the Bayesian multivariate variable selection algorithm JAM to fine-map 84 prostate cancer susceptibility loci, using summary data from a large European ancestry meta-analysis. We observe evidence for multiple independent signals at 12 regions and 99 risk signals overall. Only 15 original GWAS tag SNPs remain among the catalogue of candidate variants identified; the remainder are replaced by more likely candidates. Biological annotation of our credible set of variants indicates significant enrichment within promoter and enhancer elements, and transcription factor-binding sites, including AR, ERG and FOXA1. In 40 regions at least one variant is colocalised with an eQTL in prostate cancer tissue. The refined set of candidate variants substantially increase the proportion of familial relative risk explained by these known susceptibility regions, which highlights the importance of fine-mapping studies and has implications for clinical risk profiling.

---

#A full list of authors and their affiliations appears at the end of the paper.

Prostate cancer (PrCa) is the most common cancer among males in developed countries. As there is evidence for a large heritable component for PrCa, the identification of genetic variation that increases susceptibility may help to inform screening strategies and clinical management of patients in the future. Currently, only a handful of rare genetic variants with larger effect sizes have been reported that increase the risk of PrCa (e.g., *BRCA2* and *ATM*)[1,2]. By comparison, genome-wide association studies (GWAS) have reported >100 low-penetrance PrCa risk signals with small odds ratios (ORs)[3]. Individually, these GWAS loci only modestly influence risk. However, because the risk alleles are relatively common within the general population their cumulative impact is substantial.

When an initial GWAS identifies a susceptibility locus, any one (or more) of a large number of variants within the region may underlie the molecular mechanism that modulates risk. This includes correlated variants in linkage disequilibrium (LD) that may capture the same association signal and additional variants with independent associations. Genotyping a denser set of variants in the region facilitates characterisation of the underlying genetic architecture and makes subsequent imputation more precise and complete. Although forward stepwise selection is frequently used for fine-mapping, it has severe limitations, particularly the way LD can lead to misleading results. In this manuscript, we report the findings of a PrCa fine-mapping study in a European ancestry meta-analysis sample set that is the largest to date and utilise the well-established stochastic search and model selection framework, which more accurately represents the uncertainty in determining both the number of signals and the set of single-nucleotide polymorphisms (SNPs) that best describe the association in each region[4–7]. To leverage the large sample size from the overall meta-analysis, we use a novel multivariate Bayesian variable selection approach, which takes marginal SNP summary statistics as input and accounts for LD, to jointly analyse all SNPs in a region. We identify a catalogue of variants and further prioritise within this set through functional annotation, to assist identification of putative causal variants. This refined credible set of variants explains a substantially larger proportion of the estimated familial relative risk (FRR) of PrCa compared with the original GWAS tags.

## Results

**Replication of reported associations prior to fine-mapping.** In this study, we examined 92 PrCa GWAS risk associations within 85 distinct genomic regions reported prior to the recent meta-analysis using the OncoArray experiment[8]; due to their complexity, two regions (Chr8q24 and Chr6p21/MHC) were excluded and are subject to separate studies. Some regions contained more than one signal due to close proximity between the reported index SNPs. Summary results from the large European ancestry meta-analysis comprising 82,591 PrCa cases and 61,213 controls from eight GWAS sub-cohorts (OncoArray, iCOGS, UK stage 1 and 2, CaPS 1 and 2, BPC3 and NCI PEGASUS), imputed to the 1000 Genomes phase 3 reference panel, were used for our fine-mapping analysis.

We first assessed whether all 92 original associations had replicated with at least one variant in the region at a genome-wide significant level (marginal $P$-value $<5 \times 10^{-8}$). Five regions had not replicated and were excluded from downstream fine-mapping analyses accordingly (Supplementary Table 1). An additional 3 associations previously reported in different ancestral populations also had not replicated in our European sample set; however, these original lead variants were each situated within the region boundary of another replicated GWAS association and therefore the expanded region boundary was retained during fine-mapping

for logistical purposes, although only the associations replicated in Europeans were considered as index variants. Fine-mapping was therefore conducted for 84 replicated, previously reported GWAS signals, within 80 distinct regions (Fig. 1). This included the region encompassing the moderate penetrance risk SNP rs138213197 in *HOXB13*, which although originally identified through sequencing[9] was included due to its relatively close proximity to the GWAS association rs11650494. The *HOXB13* region therefore also served as a useful positive control during mapping, since the known causal variant exerts a relatively large effect size (OR 3.85) and has low minor allele frequency (MAF), but the signal is also detectable through a cluster of more common variants as a 'synthetic association'[10].

The eight signals that did not replicate in our European meta-analysis may remain risk loci for PrCa in other ancestral populations or specific disease phenotypes rather than overall PrCa risk, although we cannot completely exclude the possibility that some were false positives. Two of these variants were originally reported in a multi-ethnic meta-analysis (rs7153648 and rs12051443), one failed quality control (QC) due to strongly discordant MAF between individual sub-studies within the meta-analysis (rs6625711) and is also reported as having extremely discordant MAF between 1000 Genomes phase 1 and phase 3 cohorts (MAF in EUR 0.45 vs. 0.16), one was associated with young-onset disease only (rs636291), one only for aggressive PrCa (rs1571801) and the final three were reported in populations of Chinese (rs103294), Japanese (rs2055109) or African (rs7210100) ancestry and had not been confirmed in Europeans to date[11–15].

**Multivariate fine-mapping from univariate summary statistics.** We utilised Joint Analysis of Marginal summary statistics - (JAM)[16], a novel fine-mapping framework that uses summary statistics and explores multi-SNP models while accounting for LD. JAM provides inference of two important measures; (1) the most likely number of independent risk variants in the region and (2) a 95% credible set of variants that drive these signal(s). This credible set includes all variants from regression models that cumulatively reach at least 95% posterior probability in JAM's stochastic search. Prior to running JAM, the variants were pruned to eliminate high LD (initially set at $r^2 > 0.9$, decreased in $r^2 = 0.05$ increments if required, Fig. 1). JAM was run twice for each region using independent seeds of 10 million iterations each. Final credible sets for each region included the set of tag variants identified by JAM and the pruned SNPs in high LD with these tags. Region-wide Bayes factors were used to provide evidence for the minimum number of independent signals. For 75 regions JAM successfully inferred credible sets of associated variants from the meta-analysis summary statistics, with 91% concordance of variants selected between two independent runs. For the final 5 regions, JAM did not infer a strong posterior probability for any variant, therefore was unable to select candidate variants.

Overall, we identified 99 independent PrCa risk signals within the 80 replicated regions (Tables 1–3). In all, 68 regions contained a single PrCa risk association, whilst we detected evidence for multiple independent risk signals within 12 regions (15% of replicated loci). In the initial meta-analysis data set, the 80 replicated regions contained a total of 213,728 SNPs, of which 14,463 were genome-wide significant and 25,186 marginally associated with PrCa at $P < 5 \times 10^{-5}$. From this variant set, JAM identified a catalogue of 3700 SNPs as the final 95% credible set of candidate causal variants for the 75 regions successfully fine-mapped (Supplementary Data 1), whilst in the 5 regions in which JAM could not identify candidate variants, a total of 175 variants had reached genome-wide significance in the univariate meta-

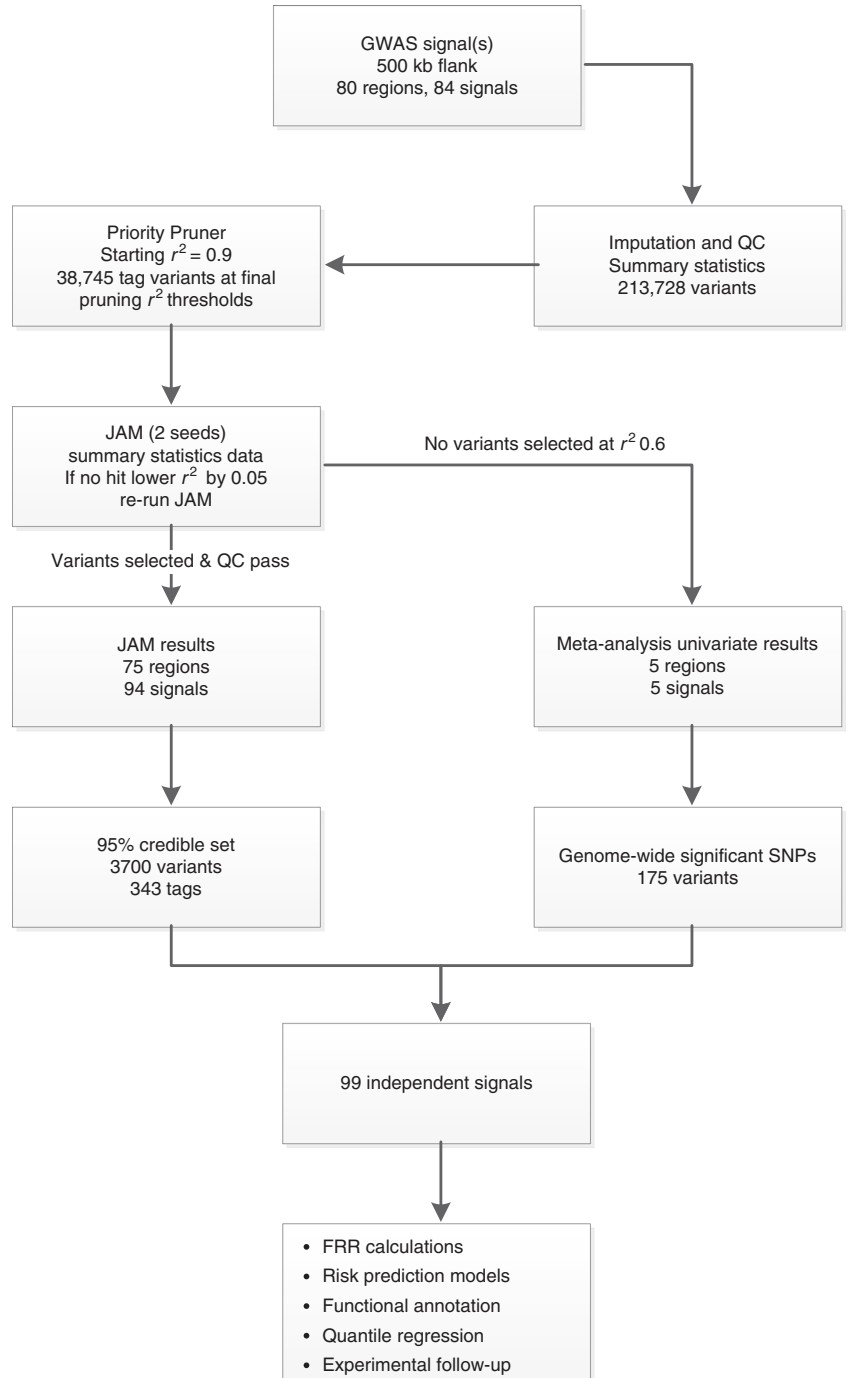

**Fig. 1** Overview of the fine-mapping workflow. Flowchart describing the procedure followed during fine-mapping, providing an overview of the outcomes at each stage and suggesting possible applications for the final catalogue of variants

GWAS results, including a novel more strongly associated lead variant in 4 of the 5 regions (Supplementary Data 1). The majority of variants within the JAM credible set were common (Supplementary Fig. 1a), with only 2 variants having MAF < 1% and 48 variants MAF < 5%; lower MAF variants do however represent the most likely candidate causal variants within certain regions. We also observed a slight increase in the distribution of univariate ORs for the novel lead variants we have identified in comparison to the original GWAS tag SNPs (Supplementary Fig. 1b). Only 15 original GWAS tag SNPs remained within the catalogue of candidate variants, with all other signals being replaced by more likely candidates. As expected, fine-mapping

performance varied by region, with 95% credible set sizes ranging from 1 to 606 variants. We did however observe strong refinement of variants within the majority of regions (median 24 variants per region overall and 21 for single-signal regions). Indeed, among the 63 single-signal regions, 30 returned a 95% credible set containing ≤20 variants, of which 20 comprised ≤10 variants and 4 returned a credible set containing a single variant. These represent the putative causal PrCa susceptibility variant within that locus and include the well-established *HOXB13* causal variant rs138213197 at Chr17q21[9], as well as rs10993994 in the promoter of *MSMB*, which modulates gene expression in prostate tissue[17–19]. These two regions serve as proof of principle; our

**Table 1 Overview of fine-mapping results by region for regions 1–27 of the 80 regions fine-mapped**

| Fine-mapping region boundary | Original index SNPs mapped | Pruning $r^2$ threshold | SNPs (tags) analysed | Number of signals | Credible set SNPs (tags) | Credible set eQTL SNPs (tags) | Credible set SNPs $P <$ 0.05 in AAs[a] | Region contribution to overall FRR of PrCa[b] |
|---|---|---|---|---|---|---|---|---|
| chr1:150158287-151158287 | rs17599629 | 0.9 | 1841 (199) | 2 | 105 (18) | 60 (10) | 29 | 0.16 (0.09, 0.24) |
| chr1:154334183-155411798 | rs1218582 | 0.9 | 1600 (309) | 1 | 2 (2) | 0 (0) | 0 | 0.12 (0.11, 0.15) |
| chr1:203991549-205018842 | rs4245739 | 0.9 | 2543 (668) | 1 | 30 (4) | 12 (2) | 5 | 0.17 (0.15, 0.20) |
| chr1:205257824-206257824 | rs1775148 | 0.6 | 2237 (325) | 1 | 0 (0) | 0 (0) | 0 | 0.07 (0.02, 0.12) |
| chr2:172809618-173915560 | rs12621278 | 0.9 | 3793 (833) | 1 | 42 (1) | 25 (1) | 26 | 0.27 (0.24, 0.31) |
| chr2:20388265-21388265 | rs13385191 | 0.9 | 2740 (716) | 1 | 6 (2) | 2 (1) | 6 | 0.13 (0.11, 0.15) |
| chr2:237940449-238943226 | rs7584330 | 0.9 | 2938 (554) | 1 | 97 (12) | 51 (11) | 17 | 0.08 (0.07, 0.10) |
| chr2:241657087-242920971 | rs3771570 | 0.9 | 2830 (479) | 3 | 14 (7) | 1 (1) | 4 | 0.65 (0.58, 0.74) |
| chr2:43053949-44137998 | rs1465618 | 0.9 | 3446 (815) | 1 | 9 (4) | 0 (0) | 0 | 0.16 (0.14, 0.18) |
| chr2:62263347-63777843 | rs721048 | 0.9 | 2323 (479) | 1 | 20 (9) | 12 (6) | 11 | 0.46 (0.41, 0.53) |
| chr2:85267735-86294297 | rs10187424 | 0.9 | 2952 (603) | 1 | 63 (6) | 31 (4) | 58 | 0.17 (0.15, 0.19) |
| chr2:9611973-10600000 | rs11902236 | 0.6 | 2961 (286) | 1 | 12 (1) | 5 (1) | 0 | 0.08 (0.02, 0.17) |
| chr2:10600001-11210730 | rs9287719 | 0.9 | 1825 (251) | 1 | 182 (2) | 1 (1) | 0 | 0.13 (0.11, 0.15) |
| chr3:112775624-113782326 | rs7611694 | 0.9 | 2392 (354) | 1 | 16 (2) | 6 (1) | 0 | 0.17 (0.15, 0.19) |
| chr3:127419046-128752313 | rs10934853 | 0.9 | 2865 (404) | 1 | 134 (10) | 17 (6) | 67 | 0.23 (0.20, 0.26) |
| chr3:140602833-141610074 | rs6763931 | 0.6 | 2054 (233) | 1 | 49 (2) | 0 (0) | 15 | 0.04 (0.01, 0.09) |
| chr3:169574517-170630102 | rs10936632 | 0.9 | 2743 (541) | 2 | 37 (4) | 0 (0) | 15 | 0.72 (0.61, 0.86) |
| chr3:86610674-87967332 | rs2660753; rs2055109[c] | 0.9 | 4020 (467) | 1 | 124 (12) | 31 (7) | 32 | 0.33 (0.29, 0.38) |
| chr4:105561534-106564626 | rs7679673 | 0.8 | 2182 (361) | 1 | 23 (2) | 0 (0) | 12 | 0.36 (0.32, 0.41) |
| chr4:73355253-74849158 | rs10009409; rs1894292 | 0.9 | 2860 (281) | 2 | 13 (3) | 5 (1) | 11 | 0.23 (0.18, 0.30) |
| chr4:95005592-96062877 | rs12500426; rs17021918 | 0.9 | 2920 (399) | 2 | 93 (9) | 24 (5) | 33 | 0.36 (0.32, 0.42) |
| chr5:172439426-173444400 | rs6869841 | 0.65 | 2407 (249) | 1 | 10 (1) | 5 (1) | 0 | 0.07 (0.02, 0.12) |
| chr5:43865545-44885415 | rs2121875 | 0.9 | 1853 (212) | 1 | 83 (3) | 0 (0) | 2 | 0.02 (0.00, 0.05) |
| chr5:780028-1600000 | rs2242652 | 0.9 | 2500 (806) | 5 | 30 (18) | 0 (0) | 11 | 2.57 (2.29, 2.93) |
| chr5:1600001-2395829 | rs12653946 | 0.9 | 4217 (1164) | 1 | 2 (2) | 0 (0) | 2 | 0.27 (0.24, 0.30) |
| chr6:10719030-11719030 | rs4713266 | 0.9 | 2500 (335) | 1 | 8 (3) | 0 (0) | 6 | 0.08 (0.07, 0.10) |
| chr6:108779211-109785189 | rs2273669 | 0.65 | 1871 (115) | 1 | 320 (3) | 134 (2) | 95 | 0.06 (0.01, 0.13) |

Published GWAS SNPs for which the signal or region replicated in our EUR meta-analysis are indicated, alongside the region co-ordinates assigned for fine-mapping analyses (GRCh37/hg19 assembly). The final priority pruner thresholds used and numbers of variants and priority pruner tags included in the analysis are shown. Summaries of the fine-mapping analysis results for each region contain the number of independent PrCa risk signals identified within each region, the size of the credible set of variants identified by JAM and the number of variants within the credible set that were also significantly associated eQTLs in TCGA PRAD data. As an additional category to assist variant prioritisation, the number of variants in the credible set that achieved a nominally significant $P$ value threshold ($P < 0.05$) in an unconnected African Ancestry GWAS is indicated. The estimated contribution of each GWAS region to the overall familial relative risk of PrCa after fine-mapping is also provided. Results for all additional regions fine-mapped are continued in Tables 2 and 3
[a] AAs African Ancestry population PrCa meta-analysis[31]
[b] 84 of the 95 original GWAS signals identified in fine-mapping replicated in our EUR meta-analysis and were used when performing calculation of Familial Relative Risk of PrCa. rs2055109, rs7210100 and rs6625711 did not replicate in EUR but are situated within the region boundaries of other replicated signals, so were not excluded prior to fine-mapping. For five previously reported variants (rs7153648, rs12051443, rs636291, rs1571801 and rs103294), no variant within the region boundary replicated in the meta-analysis, and these regions were excluded prior to Bayesian analysis
[c] Japanese signal rs2055109 did not replicate in Europeans, but is situated within the region boundary of rs2660753

methodology selected the presumed causal variants and therefore the remaining two single candidate variants are very likely to be causal and are strong candidates to test in functional studies. These two variants are an intronic SNP in *TBX1*, and a low MAF frameshift insertion in the final exon of *FAM111A*; which confirms for the first time in Europeans the GWAS hit at this locus previously reported in Japanese[11], although the European and Japanese variants are not in LD. The 12 regions with multiple independent risk signals contained 31 independent signals in total, represented by a 95% credible set of 626 variants (median 33.5 variants per region, average 20.2 variants per association signal). Prioritisation also performed well in these complex regions. In the *TERT* region at Chr5p15 we observed the highest number of independent signals, 5, and the credible set comprised only 30 SNPs. Similarly, 3 regions each containing 3 signals (Chr2q37:*FARP2*/*ANO7*, Chr17q12:*HNF1B* and Chr19q13:*KLK3*) returned a combined credible set of 61 variants representing these 9 PrCa associations. Notably, we observed that the regions found to contain multiple independent signals generally had *P*-values and marginal ORs towards the upper end of the distribution of original GWAS hits in the univariate meta-GWAS (Supplementary Fig. 2).

**Integration of annotation.** We annotated variants for indicators of putative biological functionality using data from publically available databases. Intragenic variants were ascribed to genes relative to GENCODEv19, miRNA variants using MirBasev20 and variants situated within segments of the genome under evolutionary conservation were annotated using conserved element outputs generated by four algorithms (GERP++, SiPhy Omega, SiPhy Pi and Phastcons)[20–22]. For information derived from tissue-based experimental data sets, we focused primarily on those conducted in prostate cell lines; specifically DNaseI hypersensitivity sites in three prostate cell types from seven experiments in the ENCODE project, chromatin-state characterisations by ChromHMM from Taberlay et al.[23], ChIP-seq peak locations for a variety of transcription factor (AR, CTCF, ERG, FOXA1, GABPA, GATA2, HOXB13 and NKX3.1) and histone mark (H3K27Ac, H3K27Me3 and H3K4Me3) data sets retrieved through the Cistrome Data Browser[24], and expression quantitative trait loci (eQTLs) from a set of 359 PrCa samples in the Cancer Genome Atlas (TCGA).

To formally incorporate these annotations into the prioritisation of SNPs, for the 75 regions in which JAM selected candidate variants, we investigated posterior estimates from JAM for all

**Table 2 Overview of fine-mapping results by region for regions 28–54 of the 80 regions fine-mapped**

| Fine-mapping region boundary | Original index SNPs mapped | Pruning $r^2$ threshold | SNPs (tags) analysed | Number of signals | Credible set SNPs (tags) | Credible set eQTL SNPs (tags) | Credible set SNPs $P <$ 0.05 in AAs[a] | Region contribution to overall FRR of PrCa[b] |
|---|---|---|---|---|---|---|---|---|
| chr6:116666036-117710052 | rs339331 | 0.9 | 2981 (433) | 1 | 102 (3) | 0 (0) | 101 | 0.18 (0.16, 0.20) |
| chr6:152932566-153941079 | rs1933488 | 0.9 | 3636 (599) | 1 | 86 (6) | 45 (6) | 20 | 0.12 (0.10, 0.14) |
| chr6:160081543-161382029 | rs9364554 | 0.9 | 4101 (737) | 3 | 151 (15) | 65 (10) | 7 | 1.03 (0.91, 1.19) |
| chr6:29573776-30573776 | rs7767188 | 0.75 | 7085 (464) | 1 | 606 (22) | 372 (16) | 13 | 0.07 (0.03, 0.11) |
| chr6:41036427-42043793 | rs1983891 | 0.9 | 2840 (779) | 1 | 33 (2) | 9 (2) | 33 | 0.18 (0.16, 0.21) |
| chr6:75995882-76995882 | rs9443189 | 0.6 | 1966 (72) | 1 | 0 (0) | 0 (0) | 0 | 0.06 (0.05, 0.07) |
| chr7:20494491-21496953 | rs12155172 | 0.9 | 3170 (782) | 1 | 4 (1) | 0 (0) | 2 | 0.16 (0.15, 0.19) |
| chr7:27091215-28476563 | rs10486567 | 0.9 | 3372 (691) | 1 | 11 (2) | 1 (1) | 3 | 0.34 (0.31, 0.39) |
| chr7:46937244-47937244 | rs56232506 | 0.9 | 2803 (473) | 1 | 53 (6) | 0 (0) | 34 | 0.08 (0.04, 0.13) |
| chr7:97307882-98316327 | rs6465657 | 0.9 | 2892 (411) | 1 | 31 (1) | 11 (1) | 0 | 0.27 (0.24, 0.31) |
| chr8:22938975-24028511 | rs1512268; rs2928679 | 0.9 | 3507 (755) | 2 | 74 (3) | 1 (1) | 16 | 0.77 (0.68, 0.87) |
| chr8:25392142-26410156 | rs11135910 | 0.9 | 2836 (558) | 1 | 4 (2) | 0 (0) | 0 | 0.07 (0.06, 0.09) |
| chr9:109651379-110656300 | rs817826 | 0.75 | 2817 (547) | 1 | 55 (1) | 0 (0) | 54 | 0.07 (0.04, 0.12) |
| chr9:21541998-22541998 | rs17694493 | 0.9 | 2727 (615) | 1 | 9 (3) | 0 (0) | 0 | 0.04 (0.02, 0.07) |
| chr10:103914221-104915094 | rs3850699 | 0.75 | 1802 (154) | 1 | 40 (2) | 18 (2) | 9 | 0.07 (0.03, 0.11) |
| chr10:122283141-123344709 | rs2252004 | 0.9 | 3584 (928) | 1 | 60 (7) | 0 (0) | 5 | 0.08 (0.04, 0.14) |
| chr10:126140936-127196872 | rs4962416 | 0.6 | 3150 (324) | 1 | 0 (0) | 0 (0) | 0 | 0.06 (0.02, 0.11) |
| chr10:45582985-46582985 | rs76934034 | 0.9 | 1778 (124) | 1 | 6 (2) | 2 (1) | 0 | 0.09 (0.04, 0.14) |
| chr10:51049496-52049496 | rs10993994 | 0.9 | 741 (98) | 1 | 1 (1) | 0 (0) | 1 | 1.44 (1.29, 1.64) |
| chr11:101901661-102901661 | rs11568818 | 0.9 | 2368 (453) | 1 | 2 (1) | 0 (0) | 2 | 0.17 (0.15, 0.19) |
| chr11:113307181-114307181 | rs11214775 | 0.9 | 2197 (378) | 1 | 2 (2) | 1 (1) | 1 | 0.10 (0.09, 0.12) |
| chr11:1733574-2734093 | rs7127900 | 0.9 | 2808 (781) | 1 | 40 (1) | 17 (1) | 40 | 0.66 (0.59, 0.75) |
| chr11:58415110-59610571 | rs1938781 | 0.8 | 2506 (158) | 1 | 1 (1) | 0 (0) | 0 | 0.13 (0.08, 0.18) |
| chr11:68484602-69953985 | rs7931342 | 0.9 | 4274 (990) | 2 | 44 (3) | 0 (0) | 44 | 0.85 (0.76, 0.97) |
| chr12:114185571-115584059 | rs1270884 | 0.9 | 4980 (1309) | 1 | 8 (3) | 0 (0) | 5 | 0.16 (0.14, 0.18) |
| chr12:47919618-48919618 | rs80130819 | 0.6 | 2987 (187) | 1 | 21 (2) | 0 (0) | 0 | 0.04 (0.01, 0.08) |
| chr12:49176010-50176010 | rs10875943 | 0.9 | 1641 (319) | 1 | 7 (3) | 2 (2) | 6 | 0.11 (0.09, 0.13) |

Published GWAS SNPs for which the signal or region replicated in our EUR meta-analysis are indicated, alongside the region co-ordinates assigned for fine-mapping analyses (GRCh37/hg19 assembly). The final priority pruner thresholds used and numbers of variants and priority pruner tags included in the analysis are shown. Summaries of the fine-mapping results for each region contain the number of independent PrCa risk signals identified within each region, the size of the credible set of variants identified by JAM and the number of variants within the credible set that were also significantly associated eQTLs in TCGA PRAD data. As an additional category to assist variant prioritisation, the number of variants in the credible set that achieved a nominally significant $P$ value threshold ($P < 0.05$) in an unconnected African Ancestry GWAS is indicated. The estimated contribution of each GWAS region to the overall familial relative risk of PrCa after fine-mapping is also provided. These results are a continuation from the regions displayed in Table 1 and results for all remaining regions fine-mapped are provided in Table 3
[a] AAs African Ancestry population PrCa meta-analysis[31]
[b] 84 of the 95 original GWAS signals identified for fine-mapping replicated in our EUR meta-analysis and were used when performing calculation of Familial Relative Risk of PrCa. rs2055109, rs7210100 and rs6625711 did not replicate in EUR but are situated within the region boundaries of other replicated signals, so were not excluded prior to fine-mapping. For five previously reported variants (rs7153648, rs12051443, rs636291, rs1571801 and rs103294), no variant within the region boundary replicated in the meta-analysis, and these regions were excluded prior to Bayesian analysis

37 863 pruned tags against annotation features using a conditional quantile regression (QR) analysis[25,26] at multiple quantiles (99.2, 99.4, 99.6, 99.8 and 99.95%). These correspond to posterior probabilities ranging from 0.01 to 0.99, with the exact values conditional on the linear combination of the annotations. At each quantile, we used the fitted model to calculate a predicted posterior probability given the SNP's annotation features. A single expected posterior probability was then calculated from a weighted average of these quantile-specific expected posterior probabilities with the weight reflecting both the fit (i.e., a function of the likelihood) and variance of the predicted values from the quantile-specific model to the data. We selected a single data set for each annotation category for the QR analysis to minimise correlation between variables. Whilst the majority of tag probabilities were not notably adjusted during QR, an appreciable subset of variants were up- or downgraded based upon their annotations ($\Delta$Posterior probability$_{QR}$ ranged between $-0.304$ and 0.254; 63 of the 37,863 tags had a $\Delta$Posterior probability$_{QR}$ of magnitude $\pm0.005$ or greater) (Supplementary Fig. 3). The conditional QR also facilitates identification of the annotations that demonstrate an association across the extreme quantiles of the posterior probabilities. Specifically, several annotations (eQTLs within TCGA PrCa tissue, AR and GATA2 transcription factor-binding sites, LNCaP DNase1, H3K27Ac and H3K4Me3 histone marks, enhancer and repressed chromatin states by

ChromHMM, conservation according to GERP++, higher CADD scores and protein altering variants) had statistically significant associations ($P < 1.0 \times 10^{-3}$) for at least one quantile (Supplementary Data 2). That is, the upper quantiles of the posterior probability distribution for variants with any of these annotations were larger when compared with SNPs without those annotations.

For comparison to the conditional QR approach, we also used Fisher's exact test to examine the representation of individual annotation features across variants included in the 95% credible set of prospective PrCa causal variants relative to variants not selected. Independent tests were conducted for each annotation upon the set of 37,863 tag variants analysed by JAM, of which 343 tags represented the 95% credible set of 3700 SNPs and annotations for all proxy SNPs were inherited by the tag variant. We observed significant enrichment of a number of annotations among variants in the credible set (Fig. 2, Supplementary Data 2). In particular, enrichment was found for eQTLs in the TCGA data set ($P = 1.15 \times 10^{-23}$); intragenic variants within protein-coding genes ($P = 8.15 \times 10^{-11}$; $P = 6.03 \times 10^{-5}$ for protein altering variants exclusively) but not non-coding transcripts ($P = 0.29$); promoter ($P = 1.66 \times 10^{-8}$), enhancer ($P = 3.42 \times 10^{-6}$) and transcribed ($P = 3.07 \times 10^{-7}$) ChromHMM states in prostate epithelial cells; DNaseI hypersensitivity sites from all seven ENCODE prostate data sets ($P = 1.28 \times 10^{-7}$ to $7.61 \times 10^{-17}$); for

**Table 3 Overview of fine-mapping results by region for regions 55–80 of the 80 regions fine-mapped, and summary results across all 80 regions**

| Fine-mapping region boundary | Original index SNPs mapped | Pruning $r^2$ threshold | SNPs (tags) analysed | Number of signals | Credible set SNPs (tags) | Credible set eQTL SNPs (tags) | Credible set SNPs $P <$ 0.05 in AAs[a] | Region contribution to overall FRR of PrCa[b] |
|---|---|---|---|---|---|---|---|---|
| chr12:52773904-53816821 | rs902774 | 0.9 | 3182 (553) | 1 | 28 (1) | 0 (0) | 10 | 0.32 (0.28, 0.36) |
| chr13:73228139-74468916 | rs9600079 | 0.9 | 3995 (888) | 1 | 14 (5) | 0 (0) | 10 | 0.13 (0.11, 0.14) |
| chr14:52872330-53889699 | rs8008270 | 0.9 | 2588 (410) | 1 | 12 (2) | 0 (0) | 0 | 0.11 (0.10, 0.13) |
| chr14:68502988-69626744 | rs7141529 | 0.9 | 3015 (822) | 1 | 72 (17) | 1 (1) | 4 | 0.07 (0.02, 0.12) |
| chr14:70592256-71592256 | rs8014671 | 0.6 | 2671 (139) | 1 | 0 (0) | 0 (0) | 0 | 0.05 (0.01, 0.10) |
| chr17:118965-1119162 | rs684232 | 0.9 | 3015 (848) | 1 | 11 (4) | 5 (3) | 11 | 0.21 (0.19, 0.24) |
| chr17:35547276-36603565 | rs11649743; rs4430796 | 0.9 | 1803 (444) | 3 | 26 (10) | 0 (0) | 12 | 1.24 (1.10, 1.42) |
| chr17:46302314-47211374 | rs138213197 | 0.9 | 2338 (521) | 1 | 1 (1) | 0 (0) | 0 | 6.87 (4.24, 10.41) |
| chr17:47211375-47952263 | rs11650494; rs7210100[c] | 0.9 | 1319 (378) | 1 | 83 (3) | 0 (0) | 24 | 0.07 (0.02, 0.14) |
| chr17:68608753-69617214 | rs1859962 | 0.9 | 3138 (629) | 1 | 24 (1) | 0 (0) | 20 | 0.79 (0.70, 0.89) |
| chr18:76270820-77273973 | rs7241993 | 0.9 | 3097 (488) | 1 | 3 (1) | 0 (0) | 0 | 0.16 (0.15, 0.19) |
| chr19:38235613-39244733 | rs8102476 | 0.9 | 2472 (419) | 1 | 18 (3) | 9 (2) | 16 | 0.27 (0.24, 0.31) |
| chr19:41485587-42485931 | rs11672691 | 0.9 | 2119 (337) | 1 | 4 (1) | 0 (0) | 1 | 0.19 (0.17, 0.22) |
| chr19:50840794-51864623 | rs2735839 | 0.9 | 2300 (602) | 3 | 21 (9) | 3 (1) | 8 | 0.86 (0.76, 0.98) |
| chr20:49027922-50027922 | rs12480328 | 0.9 | 1839 (309) | 1 | 44 (2) | 0 (0) | 37 | 0.08 (0.03, 0.13) |
| chr20:60515611-61515611 | rs2427345 | 0.6 | 2943 (433) | 1 | 17 (2) | 8 (2) | 0 | 0.04 (0.01, 0.09) |
| chr20:61862563-62874389 | rs6062509 | 0.9 | 3157 (831) | 1 | 21 (11) | 6 (2) | 2 | 0.16 (0.14, 0.18) |
| chr21:42401421-43401421 | rs1041449 | 0.9 | 2177 (557) | 1 | 31 (8) | 20 (6) | 7 | 0.20 (0.18, 0.23) |
| chr22:19257892-20257892 | rs2238776 | 0.9 | 2092 (373) | 1 | 1 (1) | 0 (0) | 0 | 0.08 (0.04, 0.13) |
| chr22:39952119-41297933 | rs9623117 | 0.9 | 1978 (281) | 1 | 55 (3) | 0 (0) | 6 | 0.11 (0.09, 0.13) |
| chr22:43000212-44013156 | rs5759167 | 0.9 | 3466 (781) | 2 | 18 (4) | 6 (2) | 5 | 0.76 (0.67, 0.87) |
| chrX:50741672-51741672 | rs5945619 | 0.9 | 1087 (178) | 1 | 94 (2) | 1 (1) | 93 | 1.20 (1.07, 1.37) |
| chrX:52396949-53396949 | rs2807031 | 0.6 | 493 (22) | 1 | 0 (0) | 0 (0) | 0 | 0.27 (0.11, 0.49) |
| chrX:66521550-67521550 | rs5919432 | 0.75 | 1235 (111) | 1 | 47 (1) | 0 (0) | 5 | 0.16 (0.08, 0.25) |
| chrX:69639850-70907983 | rs4844289; rs6625711[d] | 0.9 | 1274 (193) | 1 | 69 (9) | 1 (1) | 24 | 0.17 (0.16, 0.20) |
| chrX:9314135-10314135 | rs2405942 | 0.9 | 1973 (641) | 1 | 11 (5) | 1 (1) | 7 | 0.16 (0.05, 0.32) |
| **80 original GWAS loci** | **84 EUR original GWAS signals[b]** | | **213,728 (38,745)** | **99** | **3700 (343)** | **1027 (127)** | **1155** | **30.30 (26.01, 35.89)** |

Published GWAS SNPs for which the signal or region replicated in our EUR meta-analysis are indicated, alongside the region co-ordinates assigned for fine-mapping analyses (GRCh37/hg19 assembly). The final priority pruner thresholds used and numbers of variants and priority pruner tags included in the analysis are shown. Summaries of the fine-mapping analysis results for each region contain the number of independent PrCa risk signals identified within each region, the size of the credible set of variants identified by JAM and the number of variants within the credible set that were also significantly associated eQTLs in TCGA PRAD data. As an additional category to assist variant prioritisation, the number of variants in the credible set that achieved a nominally significant $P$ value threshold ($P < 0.05$) in an unconnected African Ancestry GWAS is indicated. The estimated contribution of each GWAS region to the overall familial relative risk of PrCa after fine-mapping is also provided. These results are a continuation from the regions displayed in Tables 1 and 2. Aggregated summary results across all of the 80 regions fine-mapped presented across Tables 1–3 are displayed in the final row of this table (in bold)

[a] AAs African Ancestry population PrCa meta-analysis[31]

[b] 84 of the 95 original GWAS signals identified in fine-mapping replicated in our EUR meta-analysis and were used when performing calculation of Familial Relative Risk of PrCa. rs2055109, rs7210100 and rs6625711 did not replicate in EUR but are situated within the region boundaries of other replicated signals, so were not excluded prior to fine-mapping. For five previously reported variants (rs7153648, rs12051443, rs636291, rs1571801 and rs103294), no variant within the region boundary replicated in the meta-analysis, and these regions were excluded prior to Bayesian analysis

[c] African American signal rs7210100 had MAF=0.0015, $P$=0.31 in the European meta-analysis, but is situated proximal to rs11650494

[d] SNP rs6625711 failed QC due to strongly discordant MAF between individual sub-studies within the meta-analysis and also between 1000 Genomes Phase1 and Phase3 cohorts (MAF in EUR 0.45 vs. 0.16) and is situated within the region boundary of rs4844289. Only a single signal within this region replicated in Europeans

AR ($P = 2.33 \times 10^{-15}$ to $2.86 \times 10^{-20}$), ERG ($P = 5.33 \times 10^{-12}$ to $1.00 \times 10^{-20}$), FOXA1 ($P = 9.18 \times 10^{-18}$ to $1.14 \times 10^{-18}$), GABPA ($P = 8.53 \times 10^{-12}$), GATA2 ($P = 1.24 \times 10^{-12}$), HOXB13 ($P = 8.25 \times 10^{-9}$) and NKX3.1 ($P = 9.44 \times 10^{-5}$ to $1.43 \times 10^{-15}$) transcription factor-binding sites from one or more experimental data set; for H3K27Ac ($P = 5.34 \times 10^{-19}$ to $1.39 \times 10^{-21}$) and H3K4Me3 ($P = 1.30 \times 10^{-9}$ to $8.27 \times 10^{-14}$) histone marks; and conserved elements within the human genome according to all four algorithms ($P = 1.89 \times 10^{-7}$ to $4.04 \times 10^{-11}$). Of particular interest, in over half of the regions fine-mapped, at least one variant within our credible set intersected a significantly associated eQTL with a colocalisation score >0.9 (overlap between eQTL and GWAS signal) in the TCGA PrCa data set. In all, 40 of the 75 regions contained an eQTL variant among the credible set, with 91 distinct genes represented (Tables 1–3, Supplementary Data 3). In total, 127 of the 343 tags representing the credible set

inherited an eQTL annotation (37%), compared with 5711 of the total 37,863 tags within these regions (17.8%). This corresponds to 1027 prostate eQTL variants among the 3700 credible set variants represented by the 343 JAM tags (27.8%), compared with 37,331 eQTLs from the 203,211 total variants within these 75 regions (18.4%).

Intuitively, some degree of correlation between the annotation features we examined would be expected, since regulatory regions of DNA may be indicated through various experimental techniques. Although annotations were jointly modelled in QR, any partial correlation could potentially inflate the extent of enrichment observed during independent Fisher's tests. To preclude this outcome, we examined the level of correlation between separate annotations. Correlation between replicate data sets representing the same annotation category was usually moderate to high as would be expected, with more modest levels

of correlation observed between different markers and information types (Supplementary Fig. 4). The level of correlation increased slightly when individual SNP annotations were collapsed onto tags, as the tag variants can inherit different annotations from separate SNPs. We performed logistic regression of the annotations used in the QR analysis in a single model, to evaluate their informativeness after adjustment for other annotation categories. In this regression, the TCGA eQTL, coding transcript and ERG transcription factor annotations were all highly significant after adjusting for multiple testing, whilst the AR transcription factor annotation was also nominally significant (Supplementary Fig. 5). The remaining annotations were not significant after adjustment for other annotations; however, within the range of information types selected, separate data sets represent broader or greater resolution functional information relative to one another and therefore may partially overlap with other markers whilst remaining instructive individually.

**Fine-mapping resolution**. At several regions our catalogue of variants highlighted putative biological mechanisms that may be responsible for the differential risk of PrCa development, as well as credible sets sufficiently small to enable subsequent laboratory follow-up. One example is the Chr2q37 region described by rs3771570 in the original publication[27]. The original lead variant is intronic in *FARP2*, but multiple genes are located within the region. During fine-mapping, we observed evidence for three independent signals, one more than we previously detected[28]. These signals are represented by a credible set of 14 variants from 7 tags, demonstrating highly successful refinement of the original signal (Fig. 3a, Tables 1–3, Supplementary Data 1). The majority of these prospective causal variants are centred on the *ANO7* gene, approximately 100 kb centromeric of *FARP2*. *ANO7* is expressed predominantly in the prostate (http://www.proteinatlas.org/ENSG00000146205-ANO7/tissue), unlike *FARP2*, which is ubiquitously expressed across tissue types. Within the credible set 3 tags are selected with particularly high confidence (posterior probabilities 0.72–1); all 3 represent only themselves with no additional proxy variants to consider, and are therefore the most likely causal variants underlying the 3 signals detected. Two of these 3 candidate causal variants (rs77559646 and rs77482050) are non-synonymous SNPs in *ANO7* that are uncommon among European ancestry populations, whilst the third (rs62187431) is intronic in *ANO7*. The 11 remaining variants in the credible set include one more missense SNP within *ANO7* (rs76832527), 2 intronic variants in *ANO7* (rs111770284 and rs56091437), a synonymous variant in *ANO7* (rs2074840) and 7 variants that are all intronic within other genes (*FARP2*, *PPP1R7*, *HDLBP* and *SEPT2*). Our fine-mapping results therefore strongly implicate the *ANO7* gene as a prospective biological effector modulating susceptibility for PrCa.

The region at Chr6q22 described by rs339331 in the original publication[29] presents a good example of how variant annotations can assist further prioritisation of the most likely candidate variants even within regions where the credible set remains comparatively large after fine-mapping (Fig. 3b, Tables 1–3, Supplementary Data 1). rs339331 is intronic in *RFX6*, a member of the regulatory factor X transcription factor family. We observed a single signal during fine-mapping, but due to high LD between variants the credible set comprises 102 variants from 3 tags (the top tag with posterior probability 0.76 tagging 35 proxy SNPs, another with posterior probability 0.15 tagging 40 SNPs and the last with posterior probability 0.08 tagging 27 SNPs). Only 14 of these variants demonstrate any plausible biological evidence however, therefore the credible set can be filtered to prioritise this subset of variants. Four of these are

proxies of the tag with the greatest statistical evidence, including the variant that demonstrates the greatest biological evidence for functionality; the original index SNP rs339331, which resides within a DNaseI peak, intersects binding sites for multiple transcription factors, including AR, FOXA1, GATA2, HOXB13 and NKX3.1, and is situated within a conserved element. rs339331 would therefore be ranked highest for follow-up based on combined statistical information and biological annotations, and has been demonstrated to alter *HOXB13* transcription factor binding and *RFX6* transcription during a previous functional investigation of this region[30].

At the *TMPRSS2* region on Chr21q22, we detected a single PrCa risk signal with a credible set of 31 SNPs from 8 tags, all of which are situated within the promoter region or first intron of *TMPRSS2* (Fig. 3c, Tables 1–3, Supplementary Data 1). In all, 20 of these variants are eQTLs for *TMPRSS2* in prostate tissue, whilst 2 variants intersect transcription factor-binding sites in multiple data sets, including for AR, ERG, FOXA1, GABPA, GATA2, HOXB13 and NKX3.1. In this region, the tag selected by JAM with the highest posterior probability is substantially downgraded after QR ($\Delta$Posterior probability$_{QR}$ −0.18) due to lack of overlap with informative biological annotations, therefore it and its proxies may not in fact represent the most likely candidate causal variants. An early and common event in prostate tumour development involves a translocation that forms a *TMPRSS2: ERG* fusion, bringing the *ERG* transcription factor under transcriptional control of the more active *TMPRSS2* promoter. Our fine-mapping results and biological annotations therefore allude to the possibility that subtle, heritable differences in *TMPRSS2* expression could potentially operate in conjunction with a common somatic alteration to influence development of PrCa. Intriguingly, we also observed significant enrichment for variants intersecting ERG transcription factor-binding sites among our combined credible set of candidate variants across all regions using Fisher's exact test (Supplementary Data 2, Fig. 2).

**Comparison with African Ancestry meta-analysis results**. Since LD patterns and allele frequencies of variants frequently differ among ancestral populations, as an additional prioritisation strategy we cross-checked meta-analysis results for variants in our 95% credible set against data from a meta-analysis of 10,202 cases and 10,810 controls with African Ancestry (AA)[31]. A total of 3633 of the 3700 SNPs in our credible set were available in the AA cohort, 1155 (31.8%) of which were nominally significant at $P <$ 0.05 in the AA meta-analysis. In addition, of the 175 variants that reached genome-wide significance within the five regions in which JAM did not resolve candidate variants, 111 were nominally significant in the AA data. We would hypothesise that variants demonstrating no evidence of association in the AA data set would generally represent less likely candidate causal variants than any nominally significant variants within their region specific credible set and should be assigned lower priority when considering variants for functional confirmation studies. This extra prioritisation step does not enable us to formally exclude any variants from our credible set however, as the AA analysis may be underpowered to detect association with PrCa at specific SNPs, and additional variants within the regions fine-mapped in Europeans but not included in our credible set were not examined for association in AA data.

**Estimating the GWAS loci contribution to FRR of PrCa**. The proportion of FRR of PrCa explained by these risk loci before and after fine-mapping were calculated using conditional effect estimates and standard errors derived from the OncoArray sample

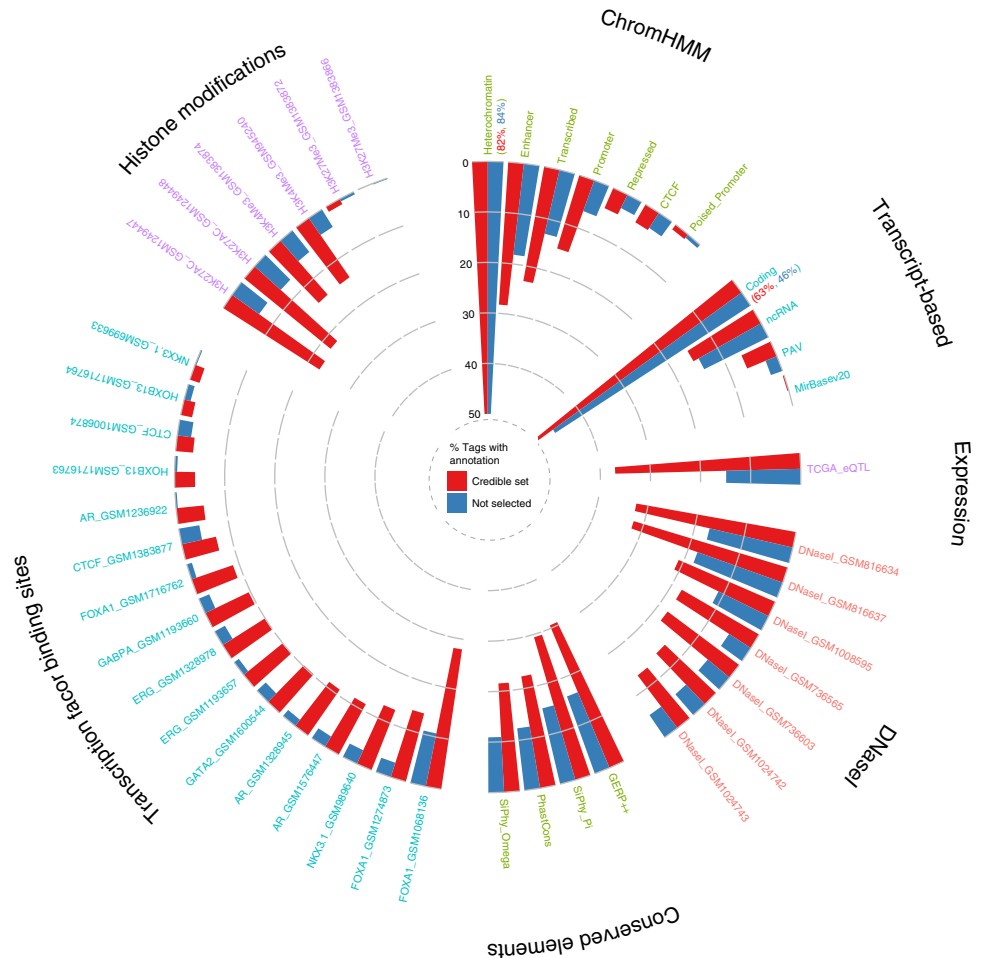

**Fig. 2** Polar bar plot depicting the proportion of tag variants assigned each functional annotation within the 95% credible set selected by JAM (red bars), relative to tags that were not selected as candidates during fine-mapping (blue bars). Binary annotations for all respective proxy variants were inherited by their tag. Annotations are grouped by category and ordered according to the proportion of variants in the credible set that receive each specific annotation. For greater clarity at lower values the plot axis is capped at 50%, therefore for annotation classes that exceed this limit (Heterochromatin and Coding) the total percentage of tags receiving the annotation is specified in brackets

sub-cohort. The post fine-mapping calculation was performed separately for the full set of 99 signals identified and a restricted subset of 84 variants (matching the number of original associations), in order to investigate the relative importance between replacement of GWAS tag SNPs and addition of extra novel signals. Single lead variants representing the independent signals were selected for this calculation. In regions containing a single signal, the JAM tag in the credible set with the highest Bayes factor was designated as the new lead variant, or for the five regions in which JAM did not resolve candidates the most strongly associated SNP in the meta-GWAS was taken instead. Within regions containing multiple independent hits, signals were represented by the combination of tags given the greatest posterior support by JAM. Our FRR calculations use conditional risk estimates incorporating uncertainty for each variant, plus a correction for potential bias due to risk estimation in the same sample as discovery and uncertainty in the specification of the FRR. This novel but more conservative method of risk calculation estimated that: (1) inclusion of only single 'best' replacement variants for each tag SNP contributes 26.5% (95% credible interval, CI, 22.7–31.5) of the known FRR of PrCa compared to 23.2% (95% CI 19.4–27.9) for the 84 previously known GWAS tag SNPs; and (2) inclusion of lead SNPs representing all of the 99 independent signals contributes 30.3% (95% CI 26.0–35.9)

(Supplementary Data 4). This substantial enhancement demonstrates that the variant catalogue identified through fine-mapping explains a greater proportion of the FRR of PrCa compared to the original GWAS index SNPs, with replacement of the 84 original GWAS tag SNPs conferring a similar magnitude of increase as addition of the 15 novel independent signals we identified. We additionally calculated the contribution to FRR of PrCa for each region individually, to highlight regions that make the greatest contributions towards PrCa susceptibility (Tables 1–3). Whilst the majority of the fine-mapped GWAS loci individually contribute a small proportion towards the FRR, six regions confer in excess of 1% each. These include the moderate penetrance *HOXB13* rs138213197 variant, which demonstrated the greatest contribution at 6.87%, and the multi-signal *TERT* locus, which explained the next highest level at 2.57%. Each of the remaining regions of higher FRR contribution contained multiple independent signals, with the exception of the single-signal *MSMB* locus. The magnitude of increase in proportion of FRR explained by each locus after fine-mapping was also generally greater for regions where additional independent signals were identified; for example, the *ANO7* region increased 6.5 fold (from 0.1% for the original GWAS tag SNP to 0.65% after fine-mapping) and the *KLK* region 1.9 fold (from 0.45 to 0.86%), partly due the identification of 2 novel signals within each.

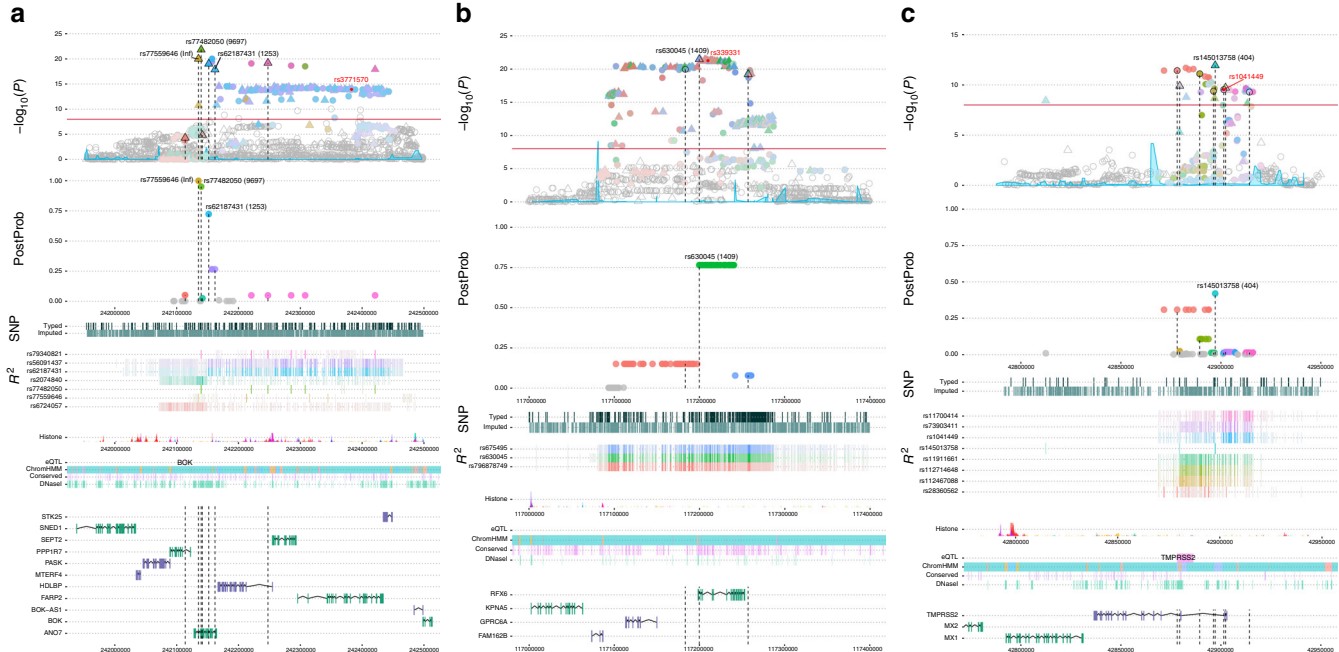

**Fig. 3** Locus Explorer plots of results and annotations at three regions. **a** Chr2q37-*ANO7*; **b** Chr6q22-*RFX6*; **c** Chr21q22-*TMPRSS2*. Upper section shows regional association plots for the initial EUR meta-analysis data depicting variant *P*-values ($-\log_{10}(P)$ panel) and fine-mapping results indicating the posterior probability of association for priority pruner tags (PostProb panel). Triangles and circles on the meta-analysis plot denote variants directly genotyped in the OncoArray study and imputed variants respectively, with colours used to indicate all variants in linkage disequilibrium (LD) at $r^2 > 0.5$ with those selected in the credible set. Names of the representative variants for each independent signal used in the familial relative risk calculation are shown in black and the original GWAS tag SNP marked in red. Only variants selected in the credible set are shown on the fine-mapping results plot, with positions of tags included in the 95% credible set marked as dashed lines and positions of all their respective proxy SNPs indicated as coloured circles. Middle section shows additional information regarding the density of directly genotyped variants within the OncoArray cohort and total imputed markers analysed (SNP panel) and the extent of variation correlated with tags in the credible set at LD $r^2 > 0.5$ ($R^2$ panel). Lower section indicates the relative positions of genes and biological annotations. Genes on the positive and negative strand are denoted by green and purple colours, respectively. Annotations displayed are as follows: histone modifications in ENCODE tier 1 cell lines (Histone track); the positions of variants that are eQTLs with prostate tumour expression in TCGA prostate adenocarcinoma samples and the respective genes for which expression is altered (eQTL track); chromatin state categorisations in the PrEC cell line by ChromHMM (ChromHMM track); the position of conserved element peaks (Conserved track); and the position of DNaseI hypersensitivity site peaks in ENCODE prostate cell lines (DNaseI track)

## Discussion

Prior to the recent OncoArray study, approximately 100 PrCa susceptibility loci identified through GWAS had been reported. Limited information was however known about the precise identity of the causal variants and functional mechanisms behind these loci despite several having been fine-mapped individually or collectively using logistic regression[28,32–35]. Here we present the largest genetic fine-mapping study for PrCa to date based on a meta-analysis of 82,591 cases and 61,213 controls of European ancestry, and employ a state-of-the-art multivariate Bayesian variable selection technique to prioritise candidate variants. We further refined results by incorporating functional annotation information using a novel QR approach, to assist prioritisation of candidate causal variants for downstream functional validation.

Since the meta-analysis comprised marginal summary effect estimates, we applied JAM, a joint Bayesian fine-mapping algorithm that accounts for LD in a multivariate analysis of univariate summary statistics, to identify credible candidate PrCa susceptibility variants. A stochastic variable selection approach provided posterior probabilities of association for each variant and combinations of variants within each region, as determined by a set of best models. This framework is preferred over alternative approaches, such as forward stepwise selection, which tend to underrepresent the uncertainty in the analysis and yield false levels of confidence for the final set of SNPs and number of signals represented by the single 'best' model. JAM also has

advantages over similar Bayesian variable selection algorithms as it incorporates an extremely computationally efficient formal reversible jump Markov Chain Monte Carlo (MCMC) stochastic model search, which allows application to very large regions and does not require a prior assumption on the maximum number of causal SNPs within each region, making it more applicable to regions with larger or unknown numbers of causal variants. Linear model-based summary data methods such as JAM represent the current state of the art and have demonstrated good performance when applied to transformed logistic ORs from binary traits as opposed to linear effects for continuous traits[36,37]. The effectiveness of logistic/linear mapping will however vary between different genomic architectures and is dependent on factors including the number of variants and correlation structure between them within each region. In general however, the approximation should work well provided no individual variants exert large effects, as expected for GWAS loci. For 5 of the 80 regions that had replicated at genome-wide significance, JAM was unable to fit a model to the summary data and consequently we could not resolve candidate variants beyond the catalogue of genome-wide significant variants within these regions. Four of these regions were not densely genotyped on the OncoArray genotyping chip, as their discovery in a multi-ethnic meta-analysis occurred only late during chip design. In addition, the top hit within these 5 regions ranked towards the weaker end of the *P*-value and effect size distributions in the univariate meta-analysis

prior to fine-mapping. The inability of JAM to resolve candidate causal variants within these regions therefore most likely results from mismatch between the reference correlation structure and meta-GWAS effect patterns, issues with the logistic/linear mapping in the presence of complex correlation structure, or possibly simply low signal to noise ratio within the data.

Use of multivariate models prioritised a 95% credible set of 3700 candidate variants from the 203,211 variants analysed within the 75 regions in which candidate variants were resolved; thereby markedly reducing the number of variants for further consideration. In addition, previous reports of multiple independent signals at several PrCa risk loci were confirmed, with evidence for multiple signals at 12 regions; of which 7 regions contained 2 signals, 4 demonstrated evidence for 3 signals and 5 signals were observed at the Chr5p15 *TERT* gene locus, which is known to contain susceptibility variants for many cancer types[38]. We observed no consistent pattern of LD relationship between the original GWAS tag SNPs and the independent signals identified through fine-mapping in the regions containing multiple independent signals (Supplementary Fig. 6). For example, at the *ANO7* locus, the original index SNP (rs3771570) is not selected in the credible set and correlated with only 1 of the 3 independent signals detected (rs62187431, $r^2 = 0.61$). In contrast, at the *TERT* region, the original index SNP (rs2242652) is in moderate or modest LD ($r^2$ 0.08–0.43) with 4 of the variants selected by JAM as representative of the 5 independent signals. Previous smaller fine-mapping studies using stepwise selection approaches had also identified evidence for independent association signals within several regions. However, these are potentially more sensitive towards subjective measures such as the *P*-value threshold chosen for secondary signal inclusion and LD level used to define the final list of candidate variants represented by the selected marker (s). Due to our substantially larger sample size and variant density available and the well-established superiority of Bayesian search procedures over stepwise selection in high-dimensional settings, we therefore consider this the most detailed fine-mapping study to date for variant prioritisation. Comparing our results to the previous iCOGS fine-mapping study[28], in which refinement of 64 GWAS loci was attempted in a smaller European ancestry cohort of 25,723 PrCa cases and 26,274 controls, 48 regions corresponding to 52 original index SNPs replicated at genome-wide significance in both studies, of which only 21 regions had been densely genotyped on the iCOGS chip (Supplementary Data 5). Within these comparable regions, 70% of the 'best candidate SNPs' established using the iCOGS sample set were also included in the credible set we have identified in this study. This indicates broad stability of the results from fine-mapping studies conducted in the same ancestral population. The additional power and more dense genotyping across all regions in this study has however facilitated further refinement of potential candidate variants, identification of additional candidate variants within several regions and refinement for the first time of a number of regions in which fine-mapping had not previously been performed or had been unsuccessful. We have confirmed the existence of multiple independent risk signals at 10 loci previously reported, including identifying extra signals at the *TERT* (Chr5p15), *ANO7* (Chr2q37) and *SLC22A3* (Chr6q25) loci, and identified multiple independent association signals for the first time at two further loci, including *KLK3* (Chr19q13). Eight regions demonstrating evidence for multiple independent signals in the iCOGS fine-mapping study were however not corroborated in this larger study. Notably, the conditional *P*-values for these secondary signals in the iCOGS fine-mapping study were below genome-wide significance in all but one of these regions. This may suggest that contrary to general assumptions that a lower burden of evidence is valid for uncorrelated variants in loci for which a

genome-wide significant association has previously been observed, instead equally stringent significance thresholds should be applied for both secondary signals and initial primary signals. It is also notable that in this well-powered study, the vast majority of regions containing multiple independent signals were first reported as associated with PrCa in early GWAS using relatively modest sample sizes. This may indicate that regions with lower effect sizes and weaker evidence for association, which require larger sample sizes for their detection, are less likely to contain additional independent risk variants. Alternatively however, it could reflect lower power for the detection of additional independently associated variants within the regions that contain weaker signals, despite the large sample cohort utilised in this study.

As would be expected, refinement of putative causal risk variants varied between regions, with credible sets ranging from a single variant or handful of variants to >100 variants for a small number of regions. The regions retaining large credible set sizes appear to result primarily from large numbers of variants in high LD with the actual causal variant as opposed to low power within the region however, rendering further refinement of these signals to facilitate functional validation studies more complicated. One approach to further prioritise candidate variants could be to leverage the different LD patterns among different ancestral populations, provided that the underlying casual variants are shared and present at sufficient frequency between populations. Cross-referencing the 3700 variants within our 95% credible set with data for an African American PrCa meta-analysis from the African Ancestry Prostate Cancer GWAS Consortium highlighted a subset of 1155 variants with nominal or genome-wide significant evidence for association in this additional population. An alternative prioritisation approach is to consider pre-existing biological information, as we have described for the *RFX6* (Chr6q22) region. We annotated variants against a number of publically available data sets, observing enrichment of several plausible markers of biological function active in prostate cell lines within our credible set, including intersection with prospective promoter and enhancer elements, DNaseI hypersensitivity sites, histone modification or transcription factor-binding peaks, and variants residing within protein-coding transcripts and conserved regions of the genome. Of particular interest, more than a quarter of the variants within our credible set were also eQTLs within the TCGA prostate adenocarcinoma data set. Given their statistical selection independent of this annotation and demonstrated effects upon gene expression, these eQTL variants should be considered high priority when selecting candidate causal variants for functional confirmation, alongside variants that modify the coding sequence of genes, or appear to reside within reliably annotated promoters or enhancers. Another important discovery of this study is that an appreciable number of highly ranked variants within the credible set are non-synonymous SNPs. This provides evidence that subtle alterations to structure and activity of specific proteins may give rise to the functional mechanisms behind a proportion of GWAS associations.

Some alternative fine-mapping algorithms integrate functional annotations during the statistical analysis when considering evidence for causality for each variant[39–42]. These methods can prove useful for enhancing variant prioritisation, provided that the annotation information is reliably indicative of causal variants. We preferred to perform statistical analysis separately from annotation and compare statistical and functional evidence for causality afterwards using conditional QR. We believe this more clearly allows the most informative annotations, and the variants that are characterised by those annotations to be highlighted within the data set, whilst also reducing the potential for

penalisation of strong candidate variants due to localised artefacts or cell line-specific effects within the whole-genome biological data sets used for annotation. Our conditional QR analysis resulted in adjustment of posterior probability for a small proportion of variants and may further assist prioritisation of the most likely functional variants among the credible set selected for each region.

Fine-mapping studies are important to reveal information on the biological mechanisms underlying disease predisposition by pinpointing potential candidate genes, signalling pathways and networks that account for differences in disease risk between individuals. In addition, these studies may help to refine the contribution of GWAS loci to PrCa risk by incorporating more likely candidate variants. This study evaluated almost all previously reported PrCa GWAS regions, apart from the highly complex Chr8q24 and major histocompatibility complex (MHC) regions and associations that did not replicate in the largest European meta-analysis to date. We then subsequently re-evaluated the contribution to FRR of these known PrCa risk loci using an enhanced method in which the overall FRR of PrCa was revised upwards from 2.0 to 2.5 to reflect the most recent estimates and we also accounted for uncertainty of various estimates that can introduce bias in these calculations. Our approach therefore provides more conservative estimates than in previous publications. We demonstrated a substantial increase in the proportion of FRR explained through fine-mapping these GWAS regions (from 23.2 to 30.3%), with detailed investigation showing that a similar proportion of this enhancement was conferred by replacement of the original tag SNPs and discovery of secondary signals. It is also noteworthy that the 7.1% magnitude of increase in FRR explained after fine-mapping known loci is substantially greater than the 4.4% increase achieved through identification of 62 novel PrCa loci[8]. This highlights the invaluable importance of fine-mapping studies for risk prediction and their potential utility in helping to inform clinical screening studies.

Fine-mapping of GWAS loci requires comprehensive examination of variation within the region. Logistical constraints generally preclude resequencing of disease-associated loci to achieve complete variant coverage in large sample cohorts and instead mandate the use of genotype array data followed by imputation, in order to achieve sufficient sample sizes. To ensure the accuracy of downstream fine-mapping analyses, stringent variant QC must be applied to imputed data, to exclude low-quality variants that may be indicative of imputation artefacts. In this study, initial pre-imputation QC of the meta-analysis data set was first performed to exclude potential genotyping errors, followed by post-imputation QC in which variants with low MAF or imputation information score, or divergent MAF consistency between dosage and 'best guess'-derived MAF estimates were excluded. The MAF estimate consistency check was performed to highlight additional variants for which reliability of imputation may be reduced and evaluation of variants excluded in this step revealed that the majority were situated within segments of the genome flagged as repetitive or otherwise ambiguous. Whilst we cannot guarantee that no causal variants at GWAS loci would be located within repetitive elements, we believe that the high proportion of variants filtered during QC that are located within potentially difficult to impute segments indicates an appropriate balance between controlling against both type I and II errors during the subsequent fine-mapping analyses. The inability to directly interrogate this category of variants during this study could however reflect a potential limitation.

The multivariate fine-mapping strategy we employed enabled identification of small numbers of prospective causal variants amenable to functional follow-up at many known PrCa susceptibility regions. Within this credible set of variants, we found

evidence of enrichment for a number of biologically plausible mechanisms through which PrCa risk could potentially be modulated. We observed multiple independent PrCa associations at 15% of the loci fine-mapped, and several candidate genes were indicated for consideration through functional annotation. As rare variants with MAF < 0.005 were not included in our analyses, we cannot exclude a contribution of rare casual variants exerting a greater effect size giving rise to synthetic associations at any GWAS loci, although our findings indicate that these are unlikely to be widespread. Importantly, replacement of the original GWAS tag SNPs with more likely candidate variants and identification of additional independent signals resulted in a substantial increase in the proportion of the FRR of PrCa explained by these loci. This finding accounts for a portion of the 'missing heritability' of PrCa and has important implications for clinical risk profiling and management of patients.

## Methods

**Identification of PrCa risk loci to fine-map.** We identified 101 independent PrCa GWAS risk associations within the literature that had been reported at genome-wide significance prior to the start of this study, the majority of which had previously been replicated within a European ancestry population[3,12]. Six of these lead variants were located within the Chr8q24 region that is associated with multiple cancer types in a highly complex manner, and three within the MHC Chr6p21 region. Due to the large numbers of variants, high levels of correlation and greater complexity within these regions, they are the subject of separate fine-mapping and risk stratification studies and were excluded from consideration in this analysis; the remaining 92 previously reported GWAS SNPs were selected for fine-mapping in this study. For 5 of these originally reported GWAS SNPs, no variant within ±500 kb replicated at genome-wide significance in our larger European meta-analysis and these loci were subsequently excluded from downstream Bayesian analyses and FRR calculations. An additional 2 GWAS SNPs originally reported in non-European ancestral populations and 1 reported in a previous meta-analysis did not replicate, but were situated <500 kb from an independent, replicated European risk association and were therefore still considered within the region boundaries of signals that were fine-mapped.

**Selection of SNPs for fine-mapping on the OncoArray.** A total of 78 PrCa risk associations that had been reported prior to the design of the OncoArray genotyping platform[43] were densely genotyped within the OncoArray sample cohort. Region boundaries for dense genotyping were defined as the greater of ±500 kb from the index SNPs or the maximum distance of any variant with $r^2 > 0.3$ to the index SNP in 1KG (phase 1 version 3, March 2012 release). All SNPs within these regions with MAF > 0.01 in any ancestral population were extracted and then we obtained Illumina Design Scores for all variants from the 1000 Genomes Project (phase I version 3, March 2012 release). From designable variants with a Design Score ≥ 0.8, we used Snagger[44] to select (a) all variants correlated with the known hits at $r^2 > 0.6$ and $P < 0.05$ in the iCOGS study, (b) all variants from lists of potentially functional variants, defined through ENCODE and RegulomeDB and (c) a set of SNPs to tag all remaining variants at $r^2 > 0.9$. The 23 risk loci reported in a recent multi-ethnic meta-analysis study[12] were not densely genotyped as these loci were reported after the OncoArray design; however, these regions were also fine-mapped in this study.

**Meta-analysis and imputation.** Genotype data for a combined 82,591 PrCa cases and 61,213 controls of European ancestry from eight GWAS (OncoArray, iCOGS, UK stage 1 and 2, CaPS 1 and 2, BPC3 and NCI PEGASUS) were used for the meta-analysis[8]. Per-allele ORs and standard errors were generated for the OncoArray and each GWAS, adjusting for principal components (PCs) and study relevant covariates using logistic regression. The OncoArray and iCOGS analyses were additionally stratified by country and study, respectively. We used the first seven PCs for OncoArray and first eight PCs for iCOGS samples, as additional components did not further reduce inflation in the test statistics. OR estimates were derived using either SNPTEST (https://mathgen.stats.ox.ac.uk/genetics_software/snptest/snptest.html) or an in-house software C++ programme. OR estimates and standard errors were combined by a fixed effects inverse variance meta-analysis using METAL[45]. All statistical tests conducted were two-sided.

IMPUTE2 was used to impute non-genotyped SNPs within a boundary flank of ±500 kb or the maximum distance of any variant with $r^2 > 0.3$ to the index SNP in 1KG phase 1 from the originally reported GWAS index SNP in the meta-analysis cohort. For the OncoArray data, un-phased imputation was carried out for all the fine-mapping regions. Where the boundaries of adjacent associations to fine-map overlapped, these were merged for imputation; therefore, imputation was performed as 82 discrete chunks. Within 3 of these chunks the separate signals to analyse were sufficiently dispersed to enable clear demarcation of the individual signals and retention of an appropriate flank distance; these 3 imputation chunks

were therefore split prior to statistical analysis and the 92 original index SNPs analysed were fine-mapped as 85 separate regions.

We conducted a two-stage post-imputation QC process. During basic QC, imputed genotype data were filtered to retain variants with INFO ≥ 0.4 and MAF ≥ 0.005. We subsequently instituted an additional QC measure to remove imputed variants with greater genotype uncertainty in which separate MAFs were calculated based on 'dosage' and 'best guess' genotypes. Large deviations between these MAF estimates for a variant would indicate unreliable imputation performance; variants for which these differed by ≥10% were excluded from analysis. An additional benefit of this methodology is that inherently it applies progressively greater stringency of QC filtering the rarer a variant is within the study population. During the post-imputation QC process, 288,033 rare variants were excluded, whilst a further 146,088 variants were removed due to low INFO score or divergent MAF consistency. This resulted in a final post-QC set for analysis of 213,728 SNPs within the 80 fine-map regions that had replicated in the initial meta-GWAS, with a minimum variant INFO score within the final data set of 0.63, and the vast majority of variants having INFO > 0.9 (Supplementary Fig. 7).

As an additional safeguard, we investigated the proportion of common variants (MAF ≥ 0.05) in 1000 Genomes European samples that were retained or excluded during our QC procedure. In total, 186,907 of 227,793 common 1000 Genomes European variants (82.1%) were included in our final post-QC data set for analysis. The vast majority of common variants excluded during QC, 37,830, were removed in the MAF consistency check step. In all, 27,070 (71.5%) of these were situated within segments of the genome flagged as repetitive or otherwise ambiguous (either masked as low complexity by RepeatMarker, or excluded by the 1000 Genomes phase 3 Strict Mask), whilst a further 4460 (11.8%) had intermediate INFO score values (0.4–0.8).

**Multivariate fine-mapping towards putative causal variants**. JAM[16] is a novel Bayesian algorithm that searches multi-SNP models in summary data by imputing the correlation structure according to a reference panel. JAM provides inference on the number of independent signals, as well as the set of potential SNPs driving those signals. Under a standard multivariate linear regression, the vector of trait values $\mathbf{y}$ are regressed on a matrix of genotypes, $\mathbf{X}$, under the following model

$$\mathbf{y} \sim N(\mathbf{X}\boldsymbol{\beta}, \mathbf{I}_N \sigma^2) \qquad (1)$$

where $\sigma^2$ represents the residual variance, $\boldsymbol{\beta}$ represents a vector of effects, which are all adjusted for one another, and $\mathbf{I}_N$ is the $N \times N$ identity matrix. Multiplying the standard model above through by the transpose of the genotype matrix, JAM makes inference under the resulting multivariate normal (MVN) model:

$$\mathbf{X}'\mathbf{y} \sim N(\mathbf{X}'\mathbf{X}\boldsymbol{\beta}, \mathbf{X}'\mathbf{X}\sigma^2) \qquad (2)$$

The motivation for using the model in (2) rather than (1) is that individual-level data are no longer required; $\mathbf{X}'\mathbf{y}$ can be derived from one-at-a-time univariate effect estimates of each variant[46,47] and $\mathbf{X}'\mathbf{X}$ from an estimate of the genetic correlation matrix. Note that in the case of the PrCa summary statistics, we derive $\mathbf{X}'\mathbf{y}$ after first mapping the univariate log ORs to approximate linear effects via their z-scores, a strategy adopted for binary traits in other linear model-based summary statistic frameworks[37,48]. Consequently, the model residuals have the same interpretation as in a linear regression of a binary outcome; they cannot exceed 1 and, under the null model, their variance $\sigma^2$ equals the trait variance, $p(1 − p)$ where $p$ is the proportion of cases. Since each region is unlikely to explain much heritability individually, we specify an inverse gamma ($\Gamma^{-1}$) prior that loosely targets the PrCa variance in the meta-GWAS:

$$\sigma^2 \sim \Gamma^{-1}(2, 0.24)$$

This corresponds to a prior expectation for $\sigma^2$ equal to the PrCa variance in the meta-GWAS, 0.24, and 95% weight over the range (0.05, 0.69). The JAM model is completed by specifying a so-called 'g-prior' over the genetic effects, $\boldsymbol{\beta}$:

$$\boldsymbol{\beta} \sim \text{MVN}(0, \tau\boldsymbol{\beta}(\mathbf{X}'\mathbf{X})^{-1})$$

The conjugate g-prior supports effects inversely proportional to the corresponding genetic co-variances and variances, as estimated from the reference matrix $\mathbf{X}'\mathbf{X}$, and has been shown to help when modelling highly correlated predictors[49]. There is a substantial literature on choices for the hyper-parameter, $\tau$; we follow recommendations to set a value equal to the maximum of $N$ and $P^2$, where $P$ is the number of variants in the region[50,51].

Crucially, both (1) and (2) are parameterised by the same vector of multivariate (i.e., correlation adjusted) effect estimates, $\boldsymbol{\beta}$; JAM is therefore able to approximate inference from a multivariate analysis of individual-level data. Optimal performance is achieved when the correlation structure $\mathbf{X}'\mathbf{X}$ is taken from the original GWAS population, rather than an external reference population. We applied JAM to summary statistics from the meta-analysis data set using LD estimated according to imputed individual-level data from the OncoArray sub-cohort of 53,449 cases and 36,225 controls in which these regions had been densely genotyped.

Similar to other Bayesian stochastic variable selection approaches, JAM models a latent vector of binary indicators, $\boldsymbol{\gamma}$, for whether each variant should be included ($\gamma_v = 1$ if variant $v$ is associated and included in the model, or 0 otherwise). Any specific configuration of indicators then specifies a specific model, $M$. Using a Bayesian stochastic search, specifically a Reversible Jump MCMC (RJMCMC) algorithm[16,52], JAM searches over different possible models. By specifying a prior on the probability of including any combination of variants, we induce a prior over the 'model space', $\boldsymbol{\gamma}$. More formally, JAM's prior over $\boldsymbol{\gamma}$ induces sparsity and accounts for the multiple testing burden through use of a 'beta-binomial' prior on the number of associated variants or variants included in any given model, which consists of a Beta distribution over the proportion of associated variants in a particular region, conditional on which prior probabilities for each possible number of associated variants follow a binomial distribution. All configurations or combinations, including the same number of variants are given identical prior probabilities. For each region we used a beta-binomial $(1, P_r)$ prior, where $P_r$ is the total number of variants in a region $r$. This places a constant prior probability for any effect in each region (i.e., one or more causal variants) of 0.5, which is split up over all possible models according to the beta-binomial distribution. Since these are previously discovered regions, this is far more generous than our prior belief would be that a random region of the genome is associated with PrCa but is more conservative for the regions in this analysis, where we estimate the false discovery rate is <10%. The marginal prior odds of any particular SNP being selected is $1/P_r$, and decreases with the total number of variants in the region, providing an intrinsic multiplicity correction as a function of region size[53–56]. The prior probabilities for ≥2, or ≥3 associated variants and so on are weakly effected by $P_r$, however, for all regions in this analysis they are equal to the second decimal place at 0.25 and 0.12, respectively (Supplementary Table 2). More detail on the JAM model and RJMCMC algorithm can be found in the original paper[16]. For this analysis, each region was analysed independently, and by running two independent JAM seeds for 10 million iterations each. The JAM output provides posterior probabilities for each variant, $\text{Pr}(\gamma_v = 1|\text{data})$, and for each combination of variants, $\text{Pr}(M = 1|\text{data})$. To determine statistical significance for individual variants, combinations of variants and for the possible number of independent signals we use Bayes factors[57], the ratio of the posterior odds to the prior odds. Specifically, we used the inference of the minimum number of independent signals in the model at a regional Bayes factor threshold of 3 to define the evidence for multiple signals.

Before running JAM, Priority Pruner v0.1.3 (http://prioritypruner.sourceforge.net) was used to LD prune the imputed meta-analysis variant set at a threshold of $r^2 = 0.9$ for the 80 regions replicated at genome-wide significance. Pruning was performed agnostic of additional prioritisation criteria (association or annotation data) to ensure unbiased Bayesian model selection. Additional pruning at lower LD levels was performed upon any regions in which the overall Bayes factor for association with PrCa fell below 1. A regional Bayes factor below 1 directly conflicts with our knowledge that these regions are robustly associated with PrCa, and was taken as an indication of collinearity; numerical instability that can occur when fitting multivariate models to highly correlated variables. Where required, the pruning threshold was lowered in $r^2 = 0.05$ increments, to a cut-off level of $r^2 = 0.6$. The pruned data set used in the final Bayesian analyses comprised a total of 38,745 selected tags.

The pruning thresholds used in the final results are listed in Tables 1–3. For each independent JAM analysis, the top models or combinations of included SNPs within each region as determined by the posterior probabilities of the models, $\text{Pr}(M = 1|\text{data})$ that summed to a cumulative posterior probability of 0.95, were used to define a run specific 95% credible set. To filter out any low confidence variants, final 95% credible sets for each region were defined according to the intersection between two independent runs of JAM, with any variants with variant-specific BF < 1 additionally removed from the amalgamated variant list due to having greater standalone evidence against association. Overall, 3761 of the 4142 unique SNPs selected by either JAM run were retained in the combined top models from both runs (90.8%), with a further 61 variants with BF < 1 removed to achieve the final 95% credible set.

**Annotation of variants for functional features**. Variants were annotated for a number of putative indicators of biological functionality or importance, using a range of publically available data sources. These annotations focussed on either the likely consequence or relevance of the variant resulting from its primary genomic context, or the proximity to annotated regulatory features within cell lines derived from normal prostate or PrCa tissues.

Gene-based annotation of variants was performed using wANNOVAR in relation to GENCODEv19 transcripts[58]. Variants residing within miRNA transcripts were subsequently added in relation to miRBase release 20 (ftp://mirbase.org/pub/mirbase/20/genomes/hsa.gff3)[59]. Annotation of variants that reside within genomic elements demonstrating evidence for evolutionary constraint was performed against conserved element peak outputs from comparative genomics analyses by four algorithms; GERP++ (http://mendel.stanford.edu/SidowLab/downloads/gerp/)[20], SiPhy_Omega, SiPhy_Pi (https://www.broadinstitute.org/scientific-community/science/projects/mammals-models/29-mammals-project-supplementary-info)[21] and PhastCons (ftp://hgdownload.cse.ucsc.edu/goldenPath/hg19/phastCons100way/)[22]. Variants were also scored for likelihood of prospective pathogenicity using CADDv1.3 (http://cadd.gs.washington.edu/score)[60].

For annotation against prospective regulatory elements within the genome, which frequently operate in a tissue-specific context, these data sets were primarily retrieved from experiments using prostate-derived cell lines. We annotated variants that intersected DNaseI peaks data in seven individual ENCODE prostate data sets from three cell lines (LNCaP, PrEC and RWPE1; GSM816637, GSM816634, GSM1008595, GSM736565, GSM736603, GSM4742 and GSM4743)[61,62]. Peak data from ChIP-seq experiments for transcription factor-binding sites and histone modifications in the LNCaP, PC3, PrEC and VCaP cell lines and human prostate tumour tissue was downloaded from the Cistrome Data Browser (http://cistrome.org/db/); a resource that accumulates publically available ChIP-seq data sets and re-analyses their raw data through a standardised pipeline and QC procedure[24]. Downloaded CistromeDB data were converted from GRCh38 to GRCh37/hg19 reference assembly co-ordinates for compatibility with our variant data set using the UCSC Genome Browser LiftOver tool (https://genome.ucsc.edu/cgi-bin/hgLiftOver). Transcription factor-binding site data were obtained for the Androgen Receptor (GSM1236922, GSM1328945 and GSM1576447), CTCF (GSM1006874 and GSM1383877), ERG (GSM1193657 and GSM1328978), FOXA1 (GSM1068136, GSM1274873 and GSM1716762), GABPA (GSM1193660), GATA2 (GSM1600544), HOXB13 (GSM1716763 and GSM1716764) and NKX3.1 (GSM699633 and GSM989640). Histone modification data were obtained for H3K27Ac (GSM1249447 and GSM1249448), H3K27me3 (GSM1383866 and GSM1383872) and H3K4me3 (GSM1383874 and GSM945240). Finally, to facilitate deeper categorisation of the genomic context of variants within prospective regulatory features, they were annotated with their chromatin state categorisations by ChromHMM from two prostate cell lines (PrEC and PC3; GSE57498), alongside three ENCODE tier 1&2 cell lines (GM12878, H1HESC and HUVEC) to enable comparison of tissue specificity for prospective regulatory elements[23,63,64].

**eQTL analysis**. Genotype and gene expression data for 494 samples with PrCa were downloaded from TCGA (https://gdc-portal.nci.nih.gov). For the genotype data set, QC was performed according to the protocol suggested by Anderson et al.[65], removing samples with heterozygosity >2 standard deviations from the mean, individuals with low genotype call rate (<95%), non-male samples and related or duplicated samples (individuals with identity-by-descent >0.185). Variants with call rate <95% were also excluded from analysis. PC analysis was performed to induce the ancestry of the TCGA samples, using the 494 TCGA samples plus 2504 samples from the 1000 Genomes Project phase 3, with non-European or Finnish samples removed from the analysis. In total, 108 samples and 106 SNPs were removed after performing QC on genotype data. For the expression data set, we observed that samples from two plates (A31K and A30D) exhibited values substantially higher than samples on the remainder of plates, therefore samples on these plates were also excluded (27 additional samples). Out of the 494 samples, 359 therefore passed QC. Genotypes for samples passing QC were subsequently imputed to the 1000 Genomes Project phase 3 reference panel within the region boundaries applied to the fine-mapping data set using IMPUTE2. In all, 227,773 variants within the fine-mapping data set passed QC thresholds in the TCGA imputed data and therefore were available for eQTL analysis. Genes with mean expression across samples of ≤6 counts or with expression variance = 0 were also excluded (4123 and 370 genes removed, respectively). Finally, expression values were quantile-normalised by samples and rank-transformed by genes. In total, 16,038 genes passed QC out of the initial 20,531.

For the eQTL analysis, 35 PEER factors[66] for the top 10,000 expressed genes were used as covariates, plus 3 genotyping PCs. eQTL analysis was performed for each region individually using FastQTL[67] with 1000 permutations and a window of 1 megabase from the transcription start site of each gene. Colocalisation tests between the eQTLs and GWAS SNPs were then performed following the approach suggested by Nica et al.[68]. First, for each significant eQTL, we added the imputed SNP to the linear regression to assess if the inclusion better explains the change in expression of the gene.

$$\text{Expression} \sim \text{genotype}(\text{eQTL}) + \text{cov} + \text{genotype}(\text{imp.SNP})$$

We retrieved the P-value of this new linear regression, assigning P-value of 1 if the eQTL and imputed SNP are the same variant. Second, we ranked the P-values in descending order for each eQTL. Finally, we calculated the colocalisation score for each pair of eQTL and imputed SNPs as:

$$\text{Colocalisation score} = (N - \text{rank})/N$$

where N is the total number of imputed SNPs in that region and rank is the rank of the imputed SNP we are including. In general, if an eQTL and an imputed SNP represent the same signal, this will be reflected by the imputed SNP having a high P-value, a low rank and consequently a high colocalisation score.

**Quantile regression**. Conditional QR across variant annotations was performed for the 75 regions successfully fine-mapped using JAM, with the 5 regions in which JAM was unable to resolve candidate variants excluded from this analysis. To minimise correlation between annotations, single data sets for each transcription factor, histone mark, DNaseI, conserved element and chromatin state by ChromHMM category were selected for investigation with conditional QR.

Specifically, the GSM736603 DNaseI, GSM1328945 AR, GSM1383877 CTCF, GSM1193657 ERG, GSM1068136 FOXA1, GSM1193660 GABPA, GSM1600544 GATA2, GSM1716763 HOXB13, GSM989640 NKX3.1, GSM1249447 H3K27Ac, GSM1383872 H3K27me3, GSM945240 H3K4me3 and GERP++ conserved element annotation fields were selected, as these were observed to be most informative for variants within the 95% credible set, whilst the PrEC cell line ChromHMM annotation was selected over the PC3 data set due to its origin from normal prostate rather than cancerous tissue. Similarly, CADD RawScore was selected, with CADD PHRED score excluded prior to the analysis. Information on whether variants were an eQTL in the TCGA data set was included. Finally, new categories were computed to ascertain whether a variant was situated within a protein-coding transcript (intronic, exonic or untranslated region), within a non-coding transcript, and whether the variant altered protein structure (non-synonymous, non-sense, frameshift or non-frameshift insertion/deletion coding variants).

All annotations were converted to binary format for the QR analysis, with the exception of CADD RawScore, which was retained as a continuous variable. Separate variables were created for each possible ChromHMM state during conversion from categorical to binary format. QR analysis was performed upon the priority pruner tag variants that were analysed by JAM, using the statistical results from those analyses. Annotations for all proxy SNPs represented by the tag variant were therefore subsequently inherited by the priority pruner tag. For the binary annotation categories, this meant that if one or more proxies had received a given annotation then the tag would also receive that annotation, whilst for the continuous CADD RawScore, the tag inherited the highest value from all associated proxies.

For a specified quantile, $\tau$, we first fit a conditional QR model to the estimated posterior probabilities from the JAM analysis for each variant. Second, we use the fitted model to calculate an expected posterior probability for each SNP given the annotation profile for that SNP. Since there is uncertainty in the choice of $\tau$, we analyse the data across a range of $\tau = (99.2, 99.4, 99.6, 99.8$ and $99.95)$ and calculate a weighted average of these expected posterior probabilities to yield a final estimate. Specifically, the posterior probability from JAM, $P$, is modelled with a conditional QR with annotation, $\mathbf{Z}$. The model is defined by an asymmetric laplace distribution (ALD):

$$P \sim N(v, \sigma^2)$$
$$v \sim \prod_i \text{ALD}(\mathbf{Z}\theta, \lambda_i, \tau_i)$$

Notice $v$ is affected by both $P$ and $\mathbf{Z}\theta$, and it suggests a weighted average of $P$ and fitted regression quantiles can approximate $v$. The density function of ALD distribution is

$$f(v|\mathbf{Z}\theta, \lambda, \tau) = \frac{\tau(1-\tau)}{\lambda} \exp\left(-\rho_\tau\left(\frac{v - \mathbf{Z}\theta}{\lambda}\right)\right)$$

For $\lambda$, the maximum is achieved at

$$\lambda^* = \frac{\sum_j \rho_\tau(v - \mathbf{Z}\theta)}{N}$$

We fix $\lambda$ to be

$$\hat{\lambda} = \frac{\sum_j \rho_\tau\left(y - \mathbf{Z}\hat{\theta}\right)}{N}$$

where $\hat{\theta}$ is coefficient estimates from classical conditional QR. With $\lambda$ fixed, only the exponential parts of the Gaussian distribution and ALD involve $v$, to which we assign weight to $P$ and $\mathbf{Z}\hat{\theta}$. Specifically classical QR yields a prediction for each SNP, $i$ as $\hat{P}_i = \mathbf{Z}\hat{\theta}_i$ at $\tau_i$; the larger the penalty $\rho_{\tau_i}\left(P_i - \hat{P}_i\right)$ and $\left(P_i - \hat{P}_i\right)^2$, the less influence $\hat{P}_i$ should have on $P_i$. We normalise the weight for $P_i$ to be 1. For $\hat{P}_i$ we assign weight

$$w_i = \exp\left(\left(-\frac{\rho_{\tau_i}\left(P_i - \hat{P}_i\right)}{\hat{\lambda}_i} - \frac{\left(P_i - \hat{P}_i\right)^2}{2\sigma^2}\right)/4\right)$$

which is approximately the penalty at $v = \left(P + \hat{P}_i\right)/2$. Our approximate value for $v$ is then

$$\hat{v} = \frac{P + \sum_i w_i Y_i}{1 + \sum_i w_i}$$

**Proportion of familial risk explained**. The contribution and comparison of the newly identified SNPs and the previously known variants to the familial risk, under

a multiplicative model, was computed using the formula

$$\sum_M (\log \lambda_m)/(\log \lambda_0)$$

Where $\lambda_0$ is the observed FRR to first degree relatives of cases and $\lambda_m$ is the FRR due to locus $m$, calculated assuming a per-allele effect:

$$\lambda_m = \frac{p_m r_m^2 + q_m}{(p_m r_m + q_m)^2}$$

where $p_m$ is the frequency of the risk allele for locus $m$, $q_m = 1 - p_m$ and $r_m$ is the estimated per-allele OR.

This calculation was performed using a Bayesian framework, which allows us to attenuate any 'winners curse' bias, and incorporates the uncertainty in estimating the variant-specific per-allele OR, $r_k$, and the value of the observed familial risk, $\lambda_0$. To correct for potential bias in effect estimation from using the same sample to determine the credible set of SNPs (the so-called 'winner's curse'), we implemented a hierarchical model similar in spirit to Zhong and Prentice[69] by placing a normal prior distribution on effect estimates of the form $\beta_m \sim N(0, \tau^2)$. Here $\beta_m$ is the log OR from the conditional model within each region, and $\tau$ is a pre-specified variance of the effect distribution reflecting our prior beliefs. For all variants, we used a conservative value of $\tau = 0.05$, reflecting a 95% prior probability density for a per-allele OR in the range of [0.91, 1.10]. For the FRR calculation, we specified a prior distribution as $\lambda_0 \sim N(2.5, 0.14^2)$, which places a 95% prior density in the range [2.22, 2.78] on the FRR of PrCa. This calculation was performed using the JAGS software[70].

To collapse our catalogue of credible variants identified through fine-mapping into a parsimonious set of SNPs matching the observed number of independent signals, we selected single representative lead variants to represent each signal. For the 63 regions in which JAM identified only a single signal, these were designated as the tag with the highest posterior evidence for association, whereas for the 5 regions in which JAM did not resolve candidate variants the variant most strongly associated with PrCa in the original meta-GWAS was designated as the novel lead variant. For the 12 regions containing multiple independent risk signals, to facilitate unbiased selection of variants representing different signals, JAM exhaustively fitted all possible multi-SNP models for the specified number of signals, and the combinations of SNPs with the highest posterior probability were selected to represent the independent signals. Separate models were run to derive the variant list for the full 99 signals identified and also a reduced set of 84 signals, matching the number of original index variants fine-mapped, to enable comparison between the contributions of replacement of the GWAS tag SNPs and addition of novel signals identified. To yield adjusted effect estimates for each lead variant in regions containing multiple signals, conditional effect estimates and standard errors for the selected 'representative' variants used for the FRR calculations were derived from the OncoArray sub-cohort of 53,449 cases and 36,225 controls, for which individual-level data were available.

**Data availability**. The meta-analysis summary data used in this fine-mapping project are available from the PRACTICAL Consortium (http://practical.icr.ac.uk/blog/?page_id=8164) or GitHub (https://github.com/oncogenetics/LocusExplorer/tree/master/Data/ProstateData). Results from the fine-mapping analyses may be explored interactively through Locus Explorer[71] (http://www.oncogenetics.icr.ac.uk/LocusExplorer/).

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

## Acknowledgements

We would particularly like to thank all the patients and control men who took part in all the studies involved in this work, as well as all the researchers, clinicians, technicians and administrative staff who have enabled this work to be carried out, and the collaborators in the PRACTICAL consortium. We wish to thank all GWAS study groups contributing to the meta-analysis data set from which these fine-mapping analyses were conducted: BPC3 (Breast and Prostate Cancer Cohort Consortium); CAPS (Cancer of the Prostate in Sweden); PEGASUS (Prostate Cancer Genome-wide Association Study of Uncommon Susceptibility Loci); The PRACTICAL (Prostate Cancer Association Group to Investigate Cancer-Associated Alterations in the Genome) Consortium; and The GAME-ON/ELLIPSE Consortium. Detailed acknowledgements and funding information for all GWAS study groups and from all the individual studies involved in the PRACTICAL Consortium are included in Supplementary Note 1.

## Author contributions

The contributions of each author to this study are as follows. Performance of fine-map analyses: T.D., E.J.S., P.J.N., E.A., D.A.L., M.N.B., C.C.B., M.M., S.Wa., Z.Z., D.V.C. and Z.K.J. Performance of variant annotation: E.J.S., E.A., M.M., C.C.B. and S.Wa. OncoArray chip design (PrCa content): T.D., E.J.S., D.A.L., A.A.O., F.R.S., X.S., P.K., F.W., Ste.C., B.E.H., D.F.E., C.A.Hai., R.A.E. and Z.K.J. Provision of DNA samples and/or phenotypic data: T.D., E.J.S., D.A.L., K.M., K.G., A.L., V.L.S., S.M.G., B.D.C., C.M.T., P.G., I.M.T., J.B., Suz.C., L.Mo., J.C., L.H., W.T., G.R., H.G., M.Al., T.N., P.Ph., N.P., J.S., T.L.T., C.Si., A.A., D.A., S.We., A.W., N.H., C.W., A.M.D., N.B., L.Mu., E.G., G.A., O.C., G.C.T., S.Ko., L.E.B., K.D.S., T.F.O., M.B., L.Ma., E.M.G., D.E.N., J.L.D., F.C.H., R.M.M., R.C.T., T.J.K., R.J.H., N.E.F., A.F., S.A.I., M.C.St., B.R., S.Ke., H.O., Y.J.L., H.W.Z., N.F., X.M., X.G., G.W., Z.S., G.G.G., M.C.So., R.J.M., L.M.F., A.S.K., B.F.D., A.V., A.G.C., L.F., R.S., M.E., M.K., J.Ll., G.C.V., K.L.P., M.S., J.Y.P., T.A.S., H.Y.L., J.L.S., C.Cy., D.W., J.Lu., E.A.O., M.S.G., B.G.N., S.F.N., M.W., R.B., M.A.R., P.I., H.B., K.C., B.H., C.M., M.L., T.S., J.K., C.J.L., E.M.J., M.R.T., P.Pa., M.C., S.L.N., L.S., Y.C.D., K.D.R., G.D.M., P.O., A.R., J.Li., S.H.T., D.W.L., L.F.N., D.L., M.G., T.K., R.K., N.U., C.Sl., V.M., M.P., S.S., F.Cl., S.J., T.Vd., S.La., P.A.T., C.A.Has., M.G.D., J.E.C., M.E.M., M.J.R., G.J., R.Hv., F.M., T.T., Y.A.K., J.X., K.T.K., L.C., H.P., A.M., A.K., S.N.T., S.K.M., D.J.S., S.Li., C.T., J.M., D.J.H., E.R., A.S., F.Ca., L.N.K., L.L.M., R.N.H., M.J.M., P.K., The PRACTICAL Consortium, F.W. and Z.K.J. Data management for meta-analysis: S.B. and X.S. Data QC for meta-analysis: T.D., E.J.S., A.A.O., F.R.S., S.I.B. and T.T. Imputation for meta-analysis: A.A.O., F.R.S., S.I.B. and L.F. Coordination of meta-analysis project: S.B., M.Ah., C.G., M.F., Ste.C., B.E.H., D.F.E., C.A.Hai., R.A.E. and Z.K.J. Provision of genotype data for individual GWAS sub-studies in meta-analysis: P.K., F.W., Ste.C., B.E.H., D.F.E., C.A.Hai., R.A.E. and Z.K.J. Writing of manuscript: E.J.S., T.D., P.J.N., D.V.C. and Z.K.J.

## Additional information

**Competing interests:** The authors declare no competing interests.

Tokhir Dadaev[1], Edward J. Saunders[1] Paul J. Newcombe[2], Ezequiel Anokian[1], Daniel A. Leongamornlert[1,3], Mark N. Brook[1], Clara Cieza-Borrella[1], Martina Mijuskovic[1], Sarah Wakerell[1], Ali Amin Al Olama[4,5], Fredrick R. Schumacher[6,7], Sonja I. Berndt[8], Sara Benlloch[1,4], Mahbubl Ahmed[1], Chee Goh[1], Xin Sheng[9], Zhuo Zhang[9], Kenneth Muir[10,11], Koveela Govindasami[1], Artitaya Lophatananon[10,11], Victoria L. Stevens[12], Susan M. Gapstur[12], Brian D. Carter[12], Catherine M. Tangen[13], Phyllis Goodman[13], Ian M. Thompson Jr.[14], Jyotsna Batra[15,16], Suzanne Chambers[17,18], Leire Moya[15,16], Judith Clements[15,16], Lisa Horvath[19,20], Wayne Tilley[21], Gail Risbridger[22,23], Henrik Gronberg[24], Markus Aly[24,25], Tobias Nordström[24,26], Paul Pharoah[4,27], Nora Pashayan[27,28], Johanna Schleutker[29,30], Teuvo L.J. Tammela[31], Csilla Sipeky[29], Anssi Auvinen[32], Demetrius Albanes[8], Stephanie Weinstein[8], Alicja Wolk[33], Niclas Hakansson[33], Catharine West[34], Alison M. Dunning[27], Neil Burnet[35], Lorelei Mucci[36], Edward Giovannucci[36], Gerald Andriole[37], Olivier Cussenot[38,39], Géraldine Cancel-Tassin[38,39], Stella Koutros[8], Laura E. Beane Freeman[8], Karina Dalsgaard Sorensen[40,41], Torben Falck Orntoft[40,41], Michael Borre[41,42], Lovise Maehle[43], Eli Marie Grindedal[43], David E. Neal[44,45,46], Jenny L. Donovan[47], Freddie C. Hamdy[46,48], Richard M. Martin[47,49,50], Ruth C. Travis[51], Tim J. Key[51], Robert J. Hamilton[52], Neil E. Fleshner[52], Antonio Finelli[52], Sue Ann Ingles[9], Mariana C. Stern[9], Barry Rosenstein[53,54], Sarah Kerns[55], Harry Ostrer[56], Yong-Jie Lu[57], Hong-Wei Zhang[58], Ninghan Feng[59], Xueying Mao[57], Xin Guo[60,61], Guomin Wang[62], Zan Sun[61], Graham G. Giles[63,64], Melissa C. Southey[65], Robert J. MacInnis[63,64], Liesel M. FitzGerald[64,66], Adam S. Kibel[67], Bettina F. Drake[37], Ana Vega[68], Antonio Gómez-Caamaño[69], Laura Fachal[4,68], Robert Szulkin[70,71], Martin Eklund[24], Manolis Kogevinas[72,73,74,75], Javier Llorca[73,76], Gemma Castaño-Vinyals[72,73,74,75], Kathryn L. Penney[77], Meir Stampfer[77], Jong Y. Park[78], Thomas A. Sellers[78], Hui-Yi Lin[79], Janet L. Stanford[80,81], Cezary Cybulski[82], Dominika Wokolorczyk[82], Jan Lubinski[82], Elaine A. Ostrander[83], Milan S. Geybels[80], Børge G. Nordestgaard[84,85], Sune F. Nielsen[84,85], Maren Weisher[85], Rasmus Bisbjerg[86], Martin Andreas Røder[87], Peter Iversen[84,87], Hermann Brenner[88,89,90], Katarina Cuk[88], Bernd Holleczek[91], Christiane Maier[92], Manuel Luedeke[92], Thomas Schnoeller[93], Jeri Kim[94], Christopher J. Logothetis[94], Esther M. John[95,96], Manuel R. Teixeira[97,98], Paula Paulo[97], Marta Cardoso[97], Susan L. Neuhausen[99], Linda Steele[99], Yuan Chun Ding[99], Kim De Ruyck[100], Gert De Meerleer[100], Piet Ost[101], Azad Razack[102], Jasmine Lim[102], Soo-Hwang Teo[103], Daniel W. Lin[80,104], Lisa F. Newcomb[80,104], Davor Lessel[105], Marija Gamulin[106], Tomislav Kulis[107], Radka Kaneva[108], Nawaid Usmani[109,110], Chavdar Slavov[111], Vanio Mitev[108], Matthew Parliament[109,110], Sandeep Singhal[109], Frank Claessens[112], Steven Joniau[113], Thomas Van den Broeck[112,113], Samantha Larkin[114], Paul A. Townsend[115], Claire Aukim-Hastie[116], Manuela Gago-Dominguez[117,118], Jose Esteban Castelao[119], Maria Elena Martinez[120], Monique J. Roobol[121], Guido Jenster[121], Ron H.N. van Schaik[122], Florence Menegaux[123], Thérèse Truong[123], Yves Akoli Koudou[123], Jianfeng Xu[124], Kay-Tee Khaw[125], Lisa Cannon-Albright[126,127], Hardev Pandha[116], Agnieszka Michael[116], Andrzej Kierzek[116], Stephen N. Thibodeau[128], Shannon K. McDonnell[129], Daniel J. Schaid[129], Sara Lindstrom[130], Constance Turman[131], Jing Ma[77], David J. Hunter[131], Elio Riboli[132], Afshan Siddiq[133], Federico Canzian[134], Laurence N. Kolonel[135], Loic Le Marchand[135], Robert N. Hoover[8], Mitchell J. Machiela[8], Peter Kraft[131], The PRACTICAL (Prostate Cancer Association Group to Investigate Cancer-Associated Alterations in the Genome) Consortium, Matthew Freedman[136], Fredrik Wiklund[24], Stephen Chanock[8], Brian E. Henderson[9], Douglas F. Easton[4,27], Christopher A. Haiman[9], Rosalind A. Eeles[1,137], David V. Conti[9] & Zsofia Kote-Jarai[1]

[1]The Institute of Cancer Research, London SW7 3RP, UK. [2]MRC Biostatistics Unit, University of Cambridge, Robinson Way, Cambridge CB2 0SR, UK. [3]Cancer Genome Project, Wellcome Trust Sanger Institute, Hinxton, Cambridge CB10 1SA, UK. [4]Centre for Cancer Genetic Epidemiology,

Department of Public Health and Primary Care, Strangeways Research Laboratory, University of Cambridge, Cambridge CB1 8RN, UK. [5]Department of Clinical Neurosciences, University of Cambridge, Cambridge CB2 0QQ, UK. [6]Department of Population and Quantitative Health Sciences, Case Western Reserve University, Cleveland, OH 44106-7219, USA. [7]Seidman Cancer Center, University Hospitals, Cleveland, OH 44106, USA. [8]Division of Cancer Epidemiology and Genetics, National Cancer Institute, NIH, Bethesda, MD 20892, USA. [9]Department of Preventive Medicine, Keck School of Medicine, University of Southern California/Norris Comprehensive Cancer Center, Los Angeles, CA 90015, USA. [10]Institute of Population Health, University of Manchester, Manchester M13 9PL, UK. [11]Warwick Medical School, University of Warwick, Coventry CV4 7AL, UK. [12]Epidemiology Research Program, American Cancer Society, 250 Williams Street, Atlanta, GA 30303, USA. [13]SWOG Statistical Center, Fred Hutchinson Cancer Research Center, Seattle, WA 98109, USA. [14]CHRISTUS Santa Rosa Hospital - Medical Center, San Antonio, TX 78229, USA. [15]Australian Prostate Cancer Research Centre-Qld, Institute of Health and Biomedical Innovation and School of Biomedical Science, Queensland University of Technology, Brisbane, QLD 4059, Australia. [16]Translational Research Institute, Brisbane, QLD 4102, Australia. [17]Menzies Health Institute Queensland, Griffith University, Gold Coast, QLD 4222, Australia. [18]Cancer Council Queensland, Fortitude Valley, QLD 4006, Australia. [19]Chris O'Brien Lifehouse (COBLH), Camperdown, Sydney, NSW 2010, Australia. [20]Garvan Institute of Medical Research, Sydney, NSW 2010, Australia. [21]Dame Roma Mitchell Cancer Research Centre, University of Adelaide, Adelaide, SA 5005, Australia. [22]Department of Anatomy and Developmental Biology, Biomedicine Discovery Institute, Monash University, Melbourne, VIC 3800, Australia. [23]Prostate Cancer Translational Research Program, Cancer Research Division, Peter MacCallum Cancer Centre, Melbourne, VIC 3000, Australia. [24]Department of Medical Epidemiology and Biostatistics, Karolinska Institute, SE-171 77 Stockholm, Sweden. [25]Department of Molecular Medicine and Surgery, Karolinska Institutet, and Department of Urology, Karolinska University Hospital, 171 76 Stockholm, Sweden. [26]Department of Clinical Sciences at Danderyd Hospital, Karolinska Institutet, 182 88 Stockholm, Sweden. [27]Centre for Cancer Genetic Epidemiology, Department of Oncology, Strangeways Laboratory, University of Cambridge, Cambridge CB1 8RN, UK. [28]Department of Applied Health Research, University College London, London WC1E 7HB, UK. [29]Institute of Biomedicine, University of Turku, FI-20014 Turku, Finland. [30]Tyks Microbiology and Genetics, Department of Medical Genetics, Turku University Hospital, 20521 Turku, Finland. [31]Department of Urology, Tampere University Hospital,  University of Tampere, Kalevantie 4, FI-33014 Tampere, Finland. [32]Department of Epidemiology, School of Health Sciences, University of Tampere, FI-33014 Tampere, Finland. [33]Division of Nutritional Epidemiology, Institute of Environmental Medicine, Karolinska Institutet, SE-171 77 Stockholm, Sweden. [34]Division of Cancer Sciences, Manchester Academic Health Science Centre, Radiotherapy Related Research, Manchester NIHR Biomedical Research Centre, The Christie Hospital NHS Foundation Trust, University of Manchester, Manchester M13 9PL, UK. [35]University of Cambridge Department of Oncology, Oncology Centre, Cambridge University Hospitals NHS Foundation Trust, Cambridge CB1 8RN, UK. [36]Department of Epidemiology, Harvard School of Public Health, Boston, MA 02115, USA. [37]Washington University School of Medicine, St. Louis, MO 63110, USA. [38]GRC N°5 ONCOTYPE-URO, UPMC Univ Paris 06, Tenon Hospital, F-75020 Paris, France. [39]CeRePP, Tenon Hospital, F-75020 Paris, France. [40]Department of Molecular Medicine, Aarhus University Hospital, 8200 Aarhus N, Denmark. [41]Department of Clinical Medicine, Aarhus University, 8200 Aarhus N, Denmark. [42]Department of Urology, Aarhus University Hospital, 8200 Aarhus N, Denmark. [43]Department of Medical Genetics, Oslo University Hospital, 0424 Oslo, Norway. [44]Department of Oncology, Addenbrooke's Hospital, University of Cambridge, Cambridge CB2 0QQ, UK. [45]Cancer Research UK Cambridge Research Institute, Li Ka Shing Centre, Cambridge CB2 0RE, UK. [46]Nuffield Department of Surgical Sciences, University of Oxford, Oxford OX1 2JD, UK. [47]School of Social and Community Medicine, University of Bristol, Canynge Hall, 39 Whatley Road, Bristol BS8 2PS, UK. [48]Faculty of Medical Science, John Radcliffe Hospital, University of Oxford, Oxford OX1 2JD, UK. [49]Medical Research Council (MRC) Integrative Epidemiology Unit, University of Bristol, Bristol BS8 2BN, UK. [50]National Institute for Health Research (NIHR) Biomedical Research Centre, University of Bristol, Bristol BS8 1TH, UK. [51]Cancer Epidemiology, Nuffield Department of Population Health, University of Oxford, Oxford OX3 7LF, UK. [52]Department of Surgical Oncology, Princess Margaret Cancer Centre, Toronto, ON M5G 2M9, Canada. [53]Department of Radiation Oncology, Icahn School of Medicine at Mount Sinai, New York, NY 10029, USA. [54]Department of Genetics and Genomic Sciences, Icahn School of Medicine at Mount Sinai, New York, NY 10029-5674, USA. [55]Department of Radiation Oncology, University of Rochester Medical Center, Rochester, NY 14620, USA. [56]Professor of Pathology and Pediatrics, Albert Einstein College of Medicine, Bronx, NY 10461, USA. [57]Centre for Molecular Oncology, Barts Cancer Institute, John Vane Science Centre, Queen Mary University of London, London EC1M 6BQ, UK. [58]Second Military Medical University, Shanghai 200433, P. R. China. [59]Wuxi Second Hospital, Nanjing Medical University, Wuxi, Jiangzhu 214003, China. [60]Department of Urology, The First Affiliated Hospital, Chongqing Medical University, Chongqing 200032, China. [61]The People's Hospital of Liaoning Province and The People's Hospital of China Medical University, Shenyang 110001, China. [62]Department of Urology, Zhongshan Hospital, Fudan University Medical College, Shanghai 200032, China. [63]Cancer Epidemiology & Intelligence Division, Cancer Council Victoria, Melbourne, VIC 3004, Australia. [64]Centre for Epidemiology and Biostatistics, Melbourne School of Population and Global Health, The University of Melbourne, Melbourne, VIC 3010, Australia. [65]Precision Medicine, School and Clinical Sciences at Monash Health, Monash University, Clayton, VIC 3168, Australia. [66]Menzies Institute for Medical Research, University of Tasmania, Hobart, TAS 7000, Australia. [67]Division of Urologic Surgery, Brigham and Womens Hospital, Boston, MA 02115, USA. [68]Fundación Pública Galega de Medicina Xenómica-SERGAS, Grupo de Medicina Xenómica, CIBERER, IDIS, Santiago de Compostela 15706, Spain. [69]Department of Radiation Oncology, Complexo Hospitalario Universitario de Santiago, SERGAS, 15706 Santiago de Compostela, Spain. [70]Division of Family Medicine, Department of Neurobiology, Care Science and Society, Karolinska Institutet, Huddinge, SE-171 77 Stockholm, Sweden. [71]Scandinavian Development Services, 182 33 Danderyd, Sweden. [72]Centre for Research in Environmental Epidemiology (CREAL), Barcelona Institute for Global Health (ISGlobal), 08003 Barcelona, Spain. [73]CIBER Epidemiología y Salud Pública (CIBERESP), 28029 Madrid, Spain. [74]IMIM (Hospital del Mar Research Institute), 08003 Barcelona, Spain. [75]Universitat Pompeu Fabra (UPF), 08002 Barcelona, Spain. [76]University of Cantabria-IDIVAL, 39005 Santander, Spain. [77]Channing Division of Network Medicine, Department of Medicine, Brigham and Women's Hospital/Harvard Medical School, Boston, MA 02184, USA. [78]Department of Cancer Epidemiology, Moffitt Cancer Center, Tampa, FL 33612, USA. [79]School of Public Health, Louisiana State University Health Sciences Center, New Orleans, LA 70112, USA. [80]Division of Public Health Sciences, Fred Hutchinson Cancer Research Center, Seattle, WA 98109-1024, USA. [81]Department of Epidemiology, School of Public Health, University of Washington, Seattle, WA 98195, USA. [82]International Hereditary Cancer Center, Department of Genetics and Pathology, Pomeranian Medical University, 70-115 Szczecin, Poland. [83]National Human Genome Research Institute, National Institutes of Health, Bethesda, MD 20892, USA. [84]Faculty of Health and Medical Sciences, University of Copenhagen, 2200 Copenhagen, Denmark. [85]Department of Clinical Biochemistry, Herlev and Gentofte Hospital, Copenhagen University Hospital, Herlev, 2200 Copenhagen, Denmark. [86]Department of Urology, Herlev and Gentofte Hospital, Copenhagen University Hospital, Herlev, 2200 Copenhagen, Denmark. [87]Copenhagen Prostate Cancer Center, Department of Urology, Rigshospitalet, Copenhagen University Hospital, DK-2730 Herlev, Denmark. [88]Division of Clinical Epidemiology and Aging Research, German Cancer Research Center (DKFZ), D-69120 Heidelberg, Germany. [89]German Cancer Consortium (DKTK), German Cancer Research Center (DKFZ), D-69120 Heidelberg, Germany. [90]Division of Preventive Oncology, German Cancer Research Center (DKFZ) and National Center for Tumor Diseases (NCT), 69120 Heidelberg, Germany. [91]Saarland Cancer Registry, 66119 Saarbrücken, Germany. [92]Institute for Human Genetics, University Hospital Ulm, 89075 Ulm, Germany. [93]Department of Urology, University Hospital Ulm, 89075 Ulm, Germany. [94]Department of Genitourinary Medical Oncology, The University of Texas MD Anderson Cancer Center, Houston, TX 77030, USA. [95]Cancer Prevention Institute

of California, Fremont, CA 94538, USA. [96]Department of Health Research & Policy (Epidemiology) and Stanford Cancer Institute, Stanford University School of Medicine, Stanford, CA 94305-5101, USA. [97]Department of Genetics, Portuguese Oncology Institute of Porto, 4200-072 Porto, Portugal. [98]Biomedical Sciences Institute (ICBAS), University of Porto, 4050-313 Porto, Portugal. [99]Department of Population Sciences, Beckman Research Institute of the City of Hope, Duarte, CA 91010, USA. [100]Ghent University, Faculty of Medicine and Health Sciences, Basic Medical Sciences, B-9000 Gent, Belgium. [101]Department of Radiotherapy, Ghent University Hospital, B-9000 Gent, Belgium. [102]Department of Surgery, Faculty of Medicine, University of Malaya, 50603 Kuala Lumpur, Malaysia. [103]Cancer Research Malaysia (CRM), Outpatient Centre, Subang Jaya Medical Centre, 47500 Subang Jaya, Selangor, Malaysia. [104]Department of Urology, University of Washington, Seattle, WA 98195, USA. [105]Institute of Human Genetics, University Medical Center Hamburg-Eppendorf, D-20246 Hamburg, Germany. [106]Division of Medical Oncology, Urogenital Unit, Department of Oncology at the University Hospital Centre Zagreb, Šalata 2, 10000 Zagreb, Croatia. [107]Department of Urology, University Hospital Center Zagreb, University of Zagreb School of Medicine, Šalata 2, 10000 Zagreb, Croatia. [108]Molecular Medicine Center, Department of Medical Chemistry and Biochemistry, Medical University of Sofia, 1431 Sofia, Bulgaria. [109]Department of Oncology, Cross Cancer Institute, University of Alberta, Edmonton, AB T6G 1Z2, Canada. [110]Division of Radiation Oncology, Cross Cancer Institute, Edmonton, AB T6G 1Z2, Canada. [111]Department of Urology and Alexandrovska University Hospital, Medical University of Sofia, 1431 Sofia, Bulgaria. [112]Molecular Endocrinology Laboratory, Department of Cellular and Molecular Medicine, KU Leuven, BE-3000 Leuven, Belgium. [113]Department of Urology, University Hospitals Leuven, BE-3000 Leuven, Belgium. [114]Southampton General Hospital, The University of Southampton, Southampton SO16 6YD, UK. [115]Manchester Cancer Research Centre, Faculty of Biology Medicine & Health, Manchester Academic Health Science Centre, NIHR Manchester Biomedical Research Centre, Health Innovation Manchester, University of Manchester, Manchester M13 9WL, UK. [116]The University of Surrey, Guildford, Surrey GU2 7XH, UK. [117]Genomic Medicine Group, Galician Foundation of Genomic Medicine, Instituto de Investigacion Sanitaria de Santiago de Compostela (IDIS), Complejo Hospitalario Universitario de Santiago, Servicio Galego de Saúde, SERGAS, 15706 Santiago de Compostela, Spain. [118]Moores Cancer Center, University of California San Diego, La Jolla, CA 92037, USA. [119]Genetic Oncology Unit, CHUVI Hospital, Complexo Hospitalario Universitario de Vigo, Instituto de Investigación Biomédica Galicia Sur (IISGS), 36204 Vigo (Pontevedra), Spain. [120]Moores Cancer Center, Department of Family Medicine and Public Health, University of California San Diego, La Jolla, CA 92093-0012, USA. [121]Department of Urology, Erasmus University Medical Center, 3015 CE Rotterdam, The Netherlands. [122]Department of Clinical Chemistry, Erasmus University Medical Center, 3015 CE Rotterdam, The Netherlands. [123]Cancer & Environment Group, Center for Research in Epidemiology and Population Health (CESP), INSERM, University Paris-Sud, University Paris-Saclay, 94807 Villejuif Cédex, France. [124]Program for Personalized Cancer Care, NorthShore University HealthSystem, Evanston, IL 60201, USA. [125]Clinical Gerontology Unit, University of Cambridge, Cambridge CB2 2QQ, UK. [126]Division of Genetic Epidemiology, Department of Medicine, University of Utah School of Medicine, Salt Lake City, UT 84112, USA. [127]George E. Wahlen Department of Veterans Affairs Medical Center, Salt Lake City, UT 84148, USA. [128]Department of Laboratory Medicine and Pathology, Mayo Clinic, Rochester, MN 55905, USA. [129]Division of Biomedical Statistics & Informatics, Mayo Clinic, Rochester, MN 55905, USA. [130]Department of Epidemiology, University of Washington, Seattle, WA 98195, USA. [131]Program in Genetic Epidemiology and Statistical Genetics, Department of Epidemiology, Harvard T.H. Chan School of Public Health, Boston, MA 02115, USA. [132]Department of Epidemiology and Biostatistics, School of Public Health, Imperial College, London SW7 2AZ, UK. [133]Genomics England, Queen Mary University of London, Dawson Hall, Charterhouse Square, London EC1M 6BQ, UK. [134]Genomic Epidemiology Group, German Cancer Research Center (DKFZ), D-69120 Heidelberg, Germany. [135]Epidemiology Program, University of Hawaii Cancer Center, Honolulu, HI 96813, USA. [136]Dana-Farber Cancer Institute, Boston, MA 02215, USA. [137]Royal Marsden NHS Foundation Trust, London SW3 6JJ, UK. These authors contributed equally: Tokhir Dadaev, Edward J. Saunders. These authors jointly supervised this work: Christopher A. Haiman, Rosalind A. Eeles, David V. Conti, Zsofia Kote-Jarai. Deceased: Brian E. Henderson. A full list of consortium members appears at the end of the paper.

## The PRACTICAL (Prostate Cancer Association Group to Investigate Cancer-Associated Alterations in the Genome) Consortium

Margaret Cook[4], Alison Thwaites[1], Michelle Guy[1], Ian Whitmore[1], Angela Morgan[1], Cyril Fisher[1], Steve Hazel[1], Naomi Livni[1], Amanda Spurdle[138], Srilakshmi Srinivasan[15,16], Mary-Anne Kedda[15,16], Joanne Aitken[17,18], Robert Gardiner[139,140], Vanessa Hayes[141], Lisa Butler[142], Renea Taylor[143], Trina Yeadon[15,16], Allison Eckert[15,16], Pamela Saunders[144], Anne-Maree Haynes[20,141], Melissa Papargiris[143], Paula Kujala[145], Kirsi Talala[146], Teemu Murtola[31,147], Kimmo Taari[148], David Dearnaley[1,137], Gill Barnett[35], Søren Bentzen[149], Rebecca Elliott[34], Hardeep Ranu[36], Belynda Hicks[150], Aurelie Vogt[150], Amy Hutchinson[151], Angela Cox[152], Michael Davis[47], Paul Brown[52], Anne George[153], Gemma Marsden[46,48], Athene Lane[47], Sarah J. Lewis[47], Clare Berry[132], Girish S. Kulkarni[52], Ants Toi[154], Andrew Evans[155], Alexandre R. Zlotta[52], Theodorus H. van der Kwast[155], Takashi Imai[156], Shiro Saito[157], Jacek Marzec[57], Guangwen Cao[58], Ji Lin[58], Jin Ling[58], Meiling Li[58], Shan-Chao Zhao[158], Guoping Ren[159], Yongwei Yu[160], Yudong Wu[161], Ji Wu[162], Bo Zhou[163], Yangling Zhang[159], Jie Li[60], Weiyang He[60], Jianming Guo[62], John Pedersen[164], John L. Hopper[64], Roger Milne[63,64], Aleksandra Klim[37], Ana Carballo[69], Ramón Lobato-Busto[165], Paula Peleteiro[69], Patricia Calvo[69], Miguel Aguado[68], José Manuel Ruiz-Dominguez[166], Lluís Cecchini[74], Lourdes Mengual[167,168], Antonio Alcaraz[169], Mariona Bustamante[72], Esther Gracia-Lavedan[72,73,74,75], Trinidad Dierssen-Sotos[73,76], Ines Gomez-Acebo[73,76], Julio Pow-Sang[170], Hyun Park[78], Babu Zachariah[78], Wojciech Kluzniak[82], Suzanne Kolb[80], Peter Klarskov[86], Christa Stegmaier[91], Walther Vogel[92], Kathleen Herkommer[171], Philipp Bohnert[93], Sofia Maia[97], Maria P. Silva[97], Sofie De Langhe[100], Hubert Thierens[100], Meng H. Tan[102], Aik T. Ong[102], Zeljko Kastelan[107], Elenko Popov[111],

Darina Kachakova[108], Atanaska Mitkova[108], Aleksandrina Vlahova[172], Tihomir Dikov[172], Svetlana Christova[172], Angel Carracedo[68,173,174], Christopher Bangma[121], F.H. Schroder[121], Sylvie Cenee[123,175], Brigitte Tretarre[176], Xavier Rebillard[177], Claire Mulot[178], Marie Sanchez[123,175], Jan Adolfsson[179,180], Par Stattin[181,182], Jan-Erik Johansson[183], Carin Cavalli-Bjoerkman[24], Ami Karlsson[116], Michael Broms[116], Huihai Wu[116], Lori Tillmans[128] & Shaun Riska[128]

[138]Molecular Cancer Epidemiology Laboratory, QIMR Berghofer Institute of Medical Research, Herston, QLD 4006, Australia. [139]School of Medicine, University of Queensland, Herston, QLD 4006, Australia. [140]Royal Brisbane & Women's Hospital, Herston, QLD 4029, Australia. [141]The Kinghorn Cancer Centre (TKCC), Victoria, NSW 2010, Australia. [142]Prostate Cancer Research Group, South Australian Health & Medical Research Institute, Adelaide, SA 5000, Australia. [143]Department of Physiology, Biomedicine Discovery Institute, Cancer Program, Monash University, Melbourne, VIC 3800, Australia. [144]University of Adelaide, North Terrace, Adelaide, SA 5005, Australia. [145]Fimlab Laboratories, Tampere University Hospital, FI-33520 Tampere, Finland. [146]Finnish Cancer Registry, FI-00130 Helsinki, Finland. [147]Faculty of Medicine and Life Sciences, University of Tampere, FI-33014 Tampere, Finland. [148]Department of Urology, Helsinki University Central Hospital and University of Helsinki, FI-00014 Helsinki, Finland. [149]Division of Biostatistics and Bioinformatics, University of Maryland Greenebaum Cancer Center, and Department of Epidemiology and Public Health, University of Maryland School of Medicine, Baltimore, MD 21201, USA. [150]Cancer Genomics Research Laboratory (CGR), Division of Cancer Epidemiology and Genetics, FNLCR Leidos Biomedical Research, National Cancer Institute, Frederick, MD 21701, USA. [151]DNA Extraction and Staging Laboratory (DESL), Cancer Genomics Research Laboratory (CGR), Division of Cancer Epidemiology and Genetics, FNLCR Leidos Biomedical Research, National Cancer Institute, Frederick, MD 21701, USA. [152]Sheffield Institute for Nucleic Acids, University of Sheffield, Sheffield S10 2TN, UK. [153]Cambridge Cancer Trials Centre, Cambridge Clinical Trials Unit - Cancer Theme, Cambridge University Hospitals NHS Foundation Trust, Cambridge CB2 0QQ, UK. [154]Department of Medical Imaging, University Health Network, Toronto, ON M5G 2C4, Canada. [155]Department of Pathology, University Health Network, Toronto, ON M5G 2C4, Canada. [156]Advanced Radiation Biology Research Program, Research Center for Charged Particle Therapy, National Institute of Radiological Sciences, Chiba 263-8555, Japan. [157]Department of Urology, National Hospital Organization Tokyo Medical Center, Tokyo 152-8902, Japan. [158]Department of Urology, Nanfang Hospital, Southern Medical University, 510515 Guangzhou, China. [159]Department of Pathology, The First Affiliated Hospital, Zhejiang University Medical College, 310009 Hangzhou, China. [160]Department of Pathology, Changhai Hospital, The Second Military Medical University, 200433 Shanghai, China. [161]Department of Urology, First Affiliated Hospital, Medical College, Zhengzhou University, 450003 Zhengzhou, China. [162]Department of Urology, North Sichuan Medical College, 637000 Nanchong, China. [163]Department of Nutrition Science, Shenyang Medical College, 110034 Shenyang, China. [164]Tissupath Pty Ltd., Melbourne, VIC 3122, Australia. [165]Department of Medical Physics, Complexo Hospitalario Universitario de Santiago, SERGAS, 15706 Santiago de Compostela, Spain. [166]Urology Department, Hospital Germans Trias I Pujol, 08916 Barcelona, Spain. [167]Laboratory and Department of Urology, Hospital Clínic, Institut d'Investigacions Biomèdiques August Pi i Sunyer (IDIBAPS), Universitat de Barcelona, 08036 Barcelona, Spain. [168]Centre de Recerca Biomèdica CELLEX, 08036 Barcelona, Spain. [169]Department and Laboratory of Urology, Hospital Clínic, Institut d'Investigacions Biomèdiques August Pi i Sunyer (IDIBAPS), Universitat de Barcelona, 08036 Barcelona, Spain. [170]Genitourinary Program, Moffitt Cancer Center, Tampa, FL 33612, USA. [171]Department of Urology, Klinikum rechts der Isar der Technischen Universitaet Muenchen, 81675 Munich, Germany. [172]Department of General and Clinical Pathology and Alexandrovska University Hospital, Medical University, 1431 Sofia, Bulgaria. [173]Center of Excellence in Genomic Medicine Research, King Abdulaziz University, Jeddah 2252 3270, Saudi Arabia. [174]Grupo de Medicina Xenómica, CIBERER, CIMUS, Universidad de Santiago de Compostela, Avenida de Barcelona, 15782 Santiago de Compostela, Spain. [175]Paris-Sud University, UMRS 1018, Cedex, 94807 Villejuif, France. [176]Hérault Cancer Registry, Montpellier cedex 5, Montpellier 34298, France. [177]Urology Department, Clinique Beau Soleil, 34070 Montpellier, France. [178]INSERM U1147, 75013 Paris, France. [179]Department of Clinical Science, Intervention and Technology, Karolinska Institutet, SE-171 77 Stockholm, Sweden. [180]Swedish Agency for Health Technology Assessment and Assessment of Social Services, SE-102 33 Stockholm, Sweden. [181]Department of Surgical and Perioperative Sciences, Urology and Andrology, Umeå University, SE-901 85 Umeå, Sweden. [182]Department of Surgical Sciences, Uppsala University, SE-751 85 Uppsala, Sweden. [183]Department of Urology, Faculty of Medicine and Health, Örebro University, SE-701 82 Örebro, Sweden.

