## [Peer Review File · Nature Communications]

Reviewers' comments:

Reviewer #1 (Remarks to the Author):

This manuscript describes a fine mapping study of SNPs found to be associated with PCa risk in previous GWAS studies. This study describes multiple significant findings comprised mainly of greatly narrowing down the number of SNPs potentially involved as causal variants. The strengths of the study include large size, very detailed and sophisticated statistical analysis, multiple novel findings, and excellent and experienced investigators.

Reviewer #2 (Remarks to the Author):

The manuscript presents the results of finemapping of 88 SNPs associated with prostate cancer. The research is thorough and very interesting. However, I have a number of queries. The manuscript would be strengthened if it were to give more general insight into the finemapping of GWAS loci and I have several queries relating to that.

(i) 64 loci have previously been finemapped by the same group on a smaller dataset. But no comparison is made between the results of the two studies. Does the current work support the findings of the previous paper, to the extent that the variants identified by the earlier work remain in the new credible sets of variants? It might be expected that increasing the sample size has increased power to detect explanatory variants within each region. So has there been an overall increase in the number of signals per region and is the number of signals in each region (in the two studies) correlated (do those regions that contained the most signals still contain the most)? This kind of insight would be of relevance for finemapping of regions associated with other traits.

(ii) Where there are multiple signals in a region how many of these are 'responsible' for the initial GWAS-association? In other words, did the initial association just pick up the strongest of the variants or a combination of several of them?

(iii) Similarly, is there any evidence that the more significant loci (which will be better powered for finemapping) contain more signals? This would indicate either that multiple signals are easier to distinguish when there is more power (so that many of the single-variant regions are simply underpowered) or that the regions that have the biggest effect on risk are also (for some reason) more likely to have multiple signals.

(iv) "Our aim was to refine all regions reported prior to the recent meta-analysis". But the recent meta-analysis, which identified a further 23 loci is from 2014. Why were these not included here?

(v) 53 of the regions were densely genotyped on the OncoArray in the majority of samples and thus would be expected to be far better imputed and be better powered to detect variants in the finemapping. Is there evidence of such a difference in the results between these 53 and the 35 that were more sparsely genotyped?

(vi) "For 65 regions JAM successfully inferred credible sets of associated variants from the meta-analysis summary statistics... Due to complex correlation structure, at the remaining 16 regions, we instead used individual level imputed genotype data from the OncoArray sub-cohort"

Firstly it's not clear why these 16 regions failed. The authors say that the success of the method depends on factors such as the correlation structure between variants in each region and that none of

the individual variants has a large effect. But what was the problem here? The authors say "we concluded the most likely explanation was divergence of the logistic/linear approximation" but it's not clear how they came to this conclusion. The failure of the method for about a fifth of the regions investigated is a particular concern, given that the JAM method is relatively new (2016). Secondly, how does the "individual level genotype data" approach applied to these 16 regions compare with using JAM? If you apply this to the regions that 'worked' for JAM do you get similar results (the same number of signals, overlapping credible sets, etc)?

(vii) "At least one variant within our credible set intersected a significantly associated eQTL variant among the credible set". This is an interesting result and the authors make a convincing case for this being higher than you would expect by chance. They sensibly use eQTLs for prostate cancer but it would be interesting to see how the results compare if they use eQTLs for other tissue types - i.e. is the enrichment for eQTLs in their credible sets apparent in multiple tissue types or just prostate cancer?

(viii) The previous prostate cancer finemapping paper stated "we estimate that these loci now explain ~38.9% of the familial relative risk of PrCa, an 8.9% improvement over the previously reported GWAS tag SNPs" but the new analysis states that "inclusion of lead SNPs representing all of the 107 independent signals contributes 27.4%". Given that this study includes more regions why has the proportion decreased?

(ix) "Those regions with large credible sets would benefit from additional finemapping". Was there any obvious reason why some regions resulted in such large credible sets, such as particularly high levels of LD or low power to distinguish signals?

(x) In the "meta-analysis and imputation" section it is stated that non-genotyped SNPs were imputed 500kb either side of the reported GWAS index SNP. This seems rather a small distance given that LD can extend a long way in some regions. Indeed, in the preceding section on the selection of SNPs for dense genotyping the authors report that they selected their dense genotyping regions as being within 500kb _or_ the maximum distance within which any variant had $r^2 > 0.3$ with the index SNP (whichever was greater). The latter seems far more sensible as it allows regions with longer-range LD to have bigger windows for finemapping. Why was this criteria not used for finemapping regions and how many regions would it have extended the region finemapped?

(xi) In the application of JAM, estimated LD between all variants must be supplied. The authors say they estimated this from their imputed data on 20,000 cases and controls. But since these data are derived from the reference panel used for imputation, why not use the imputation panel itself to estimate LD? This will avoid bias caused by variation in imputation quality between variants. Given that (I think) the 1000 Genomes data was used here, why not use something like the UK10K data, which would be even more informative?

Reviewer #3 (Remarks to the Author):

The authors present the largest fine-mapping study of prostate cancer loci in 82,591 cases and 61,213 controls of European ancestry, typed with the OncoArray or GWAS arrays, after high density imputation to the 1000 Genomes Project reference panel. They utilise a novel Bayesian fine-mapping algorithm that simultaneously dissects association signals within loci and constructs credible sets of variants driving the associations. Most loci contained a single association signal, although the maximum observed was five signals. Some loci were fine-mapped to relatively few variants. Multiple annotations were enriched for variants in the credible sets, including promoter and enhancer

elements. Improved fine-mapping after annotation pointed to specific causal genes and mechanisms in some loci.

The study has been performed well, and implements a powerful and novel approach to fine-mapping. The manuscript is well written, and the methods and results are generally clearly described.

Major comments:

1. The approach a first selecting tags, fine-mapping tags, and then folding bag in variants that are in LD with the selected tags seems sub-optimal. I assume that this is done because it becomes difficult to distinguish between variants in strong LD and the MCMC algorithm gets "stuck" at one variant, rather than visiting the space of variants in very strong LD? Could the authors take the regions in which JAM indicated a single association signal, and compare the JAM credible set with that which would be obtained from using approximate Bayes' factors (as described by Wakefield, AJHG 2007) with the meta-analysis results?
2. The authors claim that stepwise conditional analysis approaches are sub-optimal for identifying distinct association signals and subsequent fine-mapping. However, for the loci where JAM has concluded that there is a single association signal, would running a simple approximate conditional analysis in GCTA (using the same LD reference), conditioning on the lead SNP, also lead to the conclusion that there is a single causal variant (i.e. the residual association is not significant).
3. Are the results obtained from JAM consistent with other fine-mapping approaches that make use of summary data and LD references (such as FINEMAP), and what are the relative advantages of JAM?
4. How sensitive are the results of JAM to the choice of samples for the LD reference? Do the results remain the same if a different random selection of cases/controls is used?
5. Little reference is made to the allele frequency distribution of credible set variants. Are most variants (or most of the posterior probability) ascribed to common variants? Does this give any additional insight into genetic architecture?

Minor comments (specific line numbers provided):

331. Using the phrase "in a large European ancestry population" makes it sound like it is a single study in one population, rather than a meta-analysis – worth considering re-phrasing.
336. Wasn't clear how the 53+25 regions correspond to the 86 distinct regions described in line 330.
342. Would be useful to have an additional sentence with some detail about the meta-analysis (rather than the reader having to look forward to the methods).
376. Important to specify the LD reference used here (in addition to the methods).
401. Presumably a posterior probability is generated by JAM for each variant – it wasn't entirely clear where this posterior was being used, rather than the presence/absence of a variant in the credible set.
469. Not clear what is meant by "global level"?
484. Presumably, the enrichment procedure doesn't take account of the posterior probability – is there any way this could be incorporated?
711. Trans-ethnic fine-mapping – how would the variable LD structure between ethnicities be taken account of in JAM?
855. If you compare the post QC set of variants with all variants with $MAF > 0.5\%$ in 1000 Genomes Europeans – what proportion have been "successfully imputed" and carried forward for fine-mapping analysis?

Reviewers' comments:

Reviewer #1

(Remarks to the Author):

This manuscript describes a fine mapping study of SNPs found to be associated with PCa risk in previous GWAS studies. This study describes multiple significant findings comprised mainly of greatly narrowing down the number of SNPs potentially involved as causal variants. The strengths of the study include large size, very detailed and sophisticated statistical analysis, multiple novel findings, and excellent and experienced investigators.

We thank the reviewer for their validation of our endeavours.

Reviewer #2

(Remarks to the Author):

The manuscript presents the results of finemapping of 88 SNPs associated with prostate cancer. The research is thorough and very interesting. However, I have a number of queries. The manuscript would be strengthened if it were to give more general insight into the finemapping of GWAS loci and I have several queries relating to that.

(i) 64 loci have previously been finemapped by the same group on a smaller dataset. But no comparison is made between the results of the two studies. Does the current work support the findings of the previous paper, to the extent that the variants identified by the earlier work remain in the new credible sets of variants? It might be expected that increasing the sample size has increased power to detect explanatory variants within each region. So has there been an overall increase in the number of signals per region and is the number of signals in each region (in the two studies) correlated (do those regions that contained the most signals still contain the most)? This kind of insight would be of relevance for finemapping of regions associated with other traits.

The reviewer raises a valid point that the substantially increased sample size, and therefore power in this study could enable a detection of previously unidentified additional signals. Indeed, in this study we did observe evidence for the presence of an extra novel

signal in three regions previously known to contain multiple independent signals, and found evidence for multiple signals in three regions for the first time. It is also noteworthy that the vast majority of regions which contained multiple independent associations had been reported to be associated with PrCa in early GWAS with smaller sample sets, indicating that regions with weaker evidence for association or that confer lower effects upon risk are less likely to contain additional signals. We did however, also refute the presence of previously suggested multiple signals at a handful of regions, and therefore did not observe a drastic increase in the total number of signals identified with our larger cohort.

Alongside our substantially increased sample size, we have also implemented an enhanced Bayesian analysis methodology for this study that is far less sensitive to subjective measures such as the P -value threshold chosen for secondary signals and what level of LD is used to define the final list of candidate variants represented by the selected marker(s) than the stepwise selection techniques used in previous fine-mapping reports. We therefore consider the results of this study to represent the most reliable and informative PrCa susceptibility data available at present with current resources. We have however substantially expanded our general comparison between the results of this study and previous PrCa fine-mapping papers in the discussion section on page 18, in an attempt to provide the other researchers with the observations of trends that the reviewer highlights as of interest.

(ii) Where there are multiple signals in a region how many of these are 'responsible' for the initial GWAS-association? In other words, did the initial association just pick up the strongest of the variants or a combination of several of them?

We do not see a uniform relationship between the LD between the original GWAS tag hits and the novel variants selected in the credible set for the regions with multiple signals. Within certain regions the original GWAS hit(s) are strongly correlated with only one of the independent signals detected through fine-mapping, whilst for others they are in moderate LD with variants appearing to represent each of the independent signals. We have added a description of this observation to the discussion on page 18.

(iii) Similarly, is there any evidence that the more significant loci (which will be better powered for finemapping) contain more signals? This would indicate either that multiple signals are easier to distinguish when there is more power (so that many of the single-variant regions are simply underpowered) or that the regions that have the biggest effect on risk are also (for some reason) more likely to have multiple signals.

Whilst the number of loci containing multiple signals is fewer than those with a single signal, we do observe a general trend that regions in which the P -value of the original GWAS hits are more strongly significant in the marginal meta-analysis are also more likely to contain multiple signals than those in which the association is weaker. This same trend is also observed with the odds ratios of the original GWAS hits; higher odds ratios of the GWAS tag SNP are associated with a greater likelihood of the region containing multiple signals. There are however still a number of specific regions in which the association with prostate cancer is extremely strong or the odds ratio of the original GWAS hit towards the upper end of the distribution observed, yet the evidence points towards only a single signal. We have added an additional sentence explaining this observation to the 'Multivariate fine-mapping from univariate summary statistics' sub-section of the results on page 11 and have added an additional supplementary figure (Supplementary Figure 2).

(iv) "Our aim was to refine all regions reported prior to the recent meta-analysis". But the recent meta-analysis, which identified a further 23 loci is from 2014. Why were these not included here?

The meta-analysis this sentence is intended to refer to is the large European ancestry meta-analysis of 140,306 individuals that is currently under review (Schumacher *et al.*, Nature Genetics, submitted), and not the multi-ethnic meta-analysis of 87,040 individuals (Al Olama *et al.*, Nature Genetics, 2014) the reviewer mentions and the reference currently directs towards. The 23 newly reported loci from the 2014 multi-ethnic meta-analysis were indeed fine-mapped during this study, however were less densely genotyped on OncoArray due to their discovery occurring at a late stage in the design of the OncoArray genotyping chip; as described in the methods section ("The 23 risk loci reported in a recent multi-ethnic meta-analysis study¹³ were not densely genotyped as these loci were reported after the OncoArray design, however these regions were

also fine-mapped in this study.”). We thank the reviewer for drawing our attention the erroneous assignment of this reference to the previous meta-analysis and have updated this reference to relate to the appropriate study (Schumacher *et al.*, Nature Genetics, submitted).

(v) 53 of the regions were densely genotyped on the OncoArray in the majority of samples and thus would be expected to be far better imputed and be better powered to detect variants in the finemapping. Is there evidence of such a difference in the results between these 53 and the 35 that were more sparsely genotyped?

The reviewer is correct that the regions less densely genotyped on OncoArray may be less well powered to successfully identify candidate variants during fine-mapping. In our updated analysis, four out of the five regions in which JAM was unable to resolve candidate causal variants occurred in these regions. These regions generally demonstrate weaker *P*-values and odds ratios for the top hit in the marginal meta-analysis than the majority of other regions. This may reflect their genotyping coverage of the OncoArray, or it may reflect an underlying weak causal variant within these regions. It is also worth noting that we do not observe any obvious trend for larger or smaller credible set sizes in the more sparsely genotyped regions, indicating that lower effect size may be the more likely explanation. We have added a sentence to the discussion section on page 18 of the manuscript to explain that these two phenomena are likely to be the primary contributors to our inability to fine-map these 5 regions.

(vi) "For 65 regions JAM successfully inferred credible sets of associated variants from the meta-analysis summary statistics... Due to complex correlation structure, at the remaining 16 regions, we instead used individual level imputed genotype data from the OncoArray sub-cohort"

Firstly it's not clear why these 16 regions failed. The authors say that the success of the method depends on factors such as the correlation structure between variants in each region and that none of the individual variants has a large effect. But what was the problem here? The authors say "we concluded the most likely explanation was divergence of the logistic/linear approximation" but it's not clear how they came to this conclusion. The failure of the method for about a fifth of the

regions investigated is a particular concern, given that the JAM method is relatively new (2016). Secondly, how does the "individual level genotype data" approach applied to these 16 regions compare with using JAM? If you apply this to the regions that 'worked' for JAM do you get similar results (the same number of signals, overlapping credible sets, etc)?

Our inclusion of results from the R2BGLiMS analysis of individual level data was an attempt to provide the scientific community with candidate causal variants for the regions in which our primary analysis, JAM, had been unsuccessful in refining the initial association. These secondary results were however lower confidence as we had attempted to make clear within the manuscript, due to the substantially smaller sample set with genotype level data available for this analysis and the more computationally intense variant selection procedure that limited the number of iterations that we were able to model in comparison to our JAM analysis. Whilst there are a number of legitimate reasons why fine-mapping would be more prone to failure in some regions than others (in particular underlying LD structures, weaker effect sizes or lower minor allele frequencies of the causal variant), as the reviewer notes, we had been unable to refine the association approximately one fifth of regions in the primary analysis and were therefore required to adopt this hybrid approach.

In an attempt to alleviate this limitation, we have devised a modification to the JAM algorithm. Previously we were applying JAM, a linear model, directly to PrCa log-odds ratios under the assumption that the difference in effect scales (logistic vs linear) would have a negligible impact on inference. In our updated JAM algorithm, the univariate odds ratios are initially mapped to approximate linear effects via their z-scores. With this modification, we attempt to infer the effects that would have been obtained if linear regressions of PrCa were performed in the original meta-analysis; thereby providing JAM with effects on the same scale as assumed by its underlying model. This adjustment, which is used in other linear model based summary statistics frameworks, has enabled us to analyse all regions using JAM. This consequently improves our confidence in the results at regions where adoption of R2BGLiMS had been required previously, and allows a simplified and streamlined analysis procedure to be presented to the reader throughout the manuscript. Using the modified version of JAM, we were able to successfully

refine 75 out of 80 regions fine-mapped. For the small number of remaining regions where JAM was still unable to fit a model we describe the possible reasons behind this in the discussion on page 17-18.

(vii) "At least one variant within our credible set intersected a significantly associated eQTL variant among the credible set". This is an interesting result and the authors make a convincing case for this being higher than you would expect by chance. They sensibly use eQTLs for prostate cancer but it would be interesting to see how the results compare if they use eQTLs for other tissue types - i.e. is the enrichment for eQTLs in their credible sets apparent in multiple tissue types or just prostate cancer?

Establishing whether these eQTL variants operate specifically (or predominantly) in prostate tissue or ubiquitously across tissue types would be an interesting scientific question. We do however believe that it would not be feasible to examine this aspect within the scope of our manuscript, as it is a substantial undertaking to perform the eQTL analysis and hence we have conducted it for just the single most relevant tissue type for our study.

We believe that presenting the outcome of additional analyses would also substantially increase the length and reduce the conciseness of our presented results and deviate from the primary focus of our paper. We do however within this paper provide full information regarding all eQTL variants within the credible set and their associated genes in supplementary table 2a, which would enable interested parties to compare variants of interest to expression results for other tissue types from the databases or primary data of their choosing.

We are currently also supporting two separate studies (papers in preparation) that use GTEX data from a range of tissue types to perform transcriptome-wide association study (TWAS) analyses using the same initial meta-analysis data. These upcoming studies should broadly address the question of interest to the reviewer in a more appropriate and specific context.

(viii) The previous prostate cancer finemapping paper stated "we estimate that these loci now explain ~38.9% of the familial relative risk of PrCa, an 8.9% improvement over the previously

reported GWAS tag SNPs" but the new analysis states that "inclusion of lead SNPs representing all of the 107 independent signals contributes 27.4%". Given that this study includes more regions why has the proportion decreased?

This reduction in the absolute values that we report in this study for the proportion of familial relative risk (FRR) explained result from three aspects of an improved methodology that we have implemented in this paper. Firstly, we have revised upwards the estimate of the FRR of PrCa that was used in previous studies from 2.0 to 2.5 in line with the most recent estimates, which in turn has led to a substantial decrease in the proportion that is explained by the known GWAS variants in comparison to previous studies that used the lower FRR estimate. Secondly, we have implemented an enhanced methodology for calculating the FRR that accounts for potential bias resulting from uncertainty in the conditional risk estimates for each variant, due to risk estimation in the same sample as discovery and uncertainty in the specification of the familial relative risk. The latter factor in particular results in more conservative estimates, due to mitigation of the "winner's curse" effect. Thirdly, we also use conditional effect estimates for variants in regions with multiple independent associations for this calculation, whereas unadjusted effect estimates were employed in previous studies.

The primary goal of the calculation of the proportion of FRR explained within our paper is to demonstrate the relative improvement achieved through fine-mapping towards the likely causal variants, as opposed to the absolute estimates themselves. To avoid reader confusion in comparison to previous publications, we have however added additional explanation of the enhancements in the methodology we have applied to the discussion on page 20. We also refer to the companion meta-analysis paper in which this methodology will be applied for the first time, and which forms the new baseline estimate for the proportion of FRR explained by the original GWAS tag hits.

(ix) "Those regions with large credible sets would benefit from additional finemapping". Was there any obvious reason why some regions resulted in such large credible sets, such as particularly high levels of LD or low power to distinguish signals?

The primary reason that some regions returned a larger credible set appears to relate to linkage disequilibrium patterns, specifically large numbers of correlated variants around the signal identified. For regions with a credible set >50 variants (22 regions, 2,761 variants in the credible set represented by 172 tags), the mean number of variants per tag selected in the analysis is 16.1, whereas for regions with a credible set ≤50 variants (53 regions, 939 variants represented by 171 tags) the mean number of variants per tag is just 5.5. We have added a sentence to the discussion on page 19 to summarise this observation.

(x) In the "meta-analysis and imputation" section it is stated that non-genotyped SNPs were imputed 500kb either side of the reported GWAs index SNP. This seems rather a small distance given that LD can extend a long way in some regions. Indeed, in the preceding section on the selection of SNPs for dense genotyping the authors report that they selected their dense genotyping regions as being within 500kb or the maximum distance within which any variant had $r^2 > 0.3$ with the index SNP (whichever was greater). The latter seems far more sensible as it allows regions with longer-range LD to have bigger windows for finemapping. Why was this criteria not used for finemapping regions and in how many regions would it have extended the region finemapped?

The additional $r^2 > 0.3$ criteria with the original GWAS index SNP was applied for the imputation step as well as for the selection of SNPs for dense genotyping as a safety measure against regions in which LD might extend for long distances. We did not in practice however observe long distance LD beyond the nominal 500kb flank for any region. For simplicity we therefore generally refer to the region boundaries in relation to the standard +/-500kb flank and had omitted to mention the extra LD criteria at this section of the methods. We have now added this extra information to the 'Meta-analysis and Imputation' section of the methods on page 22 to demonstrate clearly to the reader that LD with the original GWAS hits does not extend beyond any region boundaries analysed in this study.

(xi) In the application of JAM, estimated LD between all variants must be supplied. The authors say they estimated this from their imputed data on 20,000 cases and controls. But since

these data are derived from the reference panel used for imputation, why not use the imputation panel itself to estimate LD? This will avoid bias caused by variation in imputation quality between variants. Given that (I think) the 1000 Genomes data was used here, why not use something like the UK10K data, which would be even more informative?

To clarify, we estimated LD using individual level data from the entire OncoArray sub-cohort of almost 90,000 samples. Whilst some variants are indeed imputed according to the 1000 Genomes reference panel, it should be kept in mind that a large proportion were also directly genotyped. Ideally, fine-mapping methods should estimate correlations from the actual samples analysed. Since we were in the unusual situation of having access to individual genotypes for a large cohort of study participants including a high proportion of directly genotyped variants, we therefore preferred to use this data for LD estimation, rather than an external dataset such as the UK10K. In addition, at the time of imputation and analysis in the main meta-analysis, the UK10K data was not readily available to our consortium.

Reviewer #3

(Remarks to the Author):

The authors present the largest fine-mapping study of prostate cancer loci in 82,591 cases and 61,213 controls of European ancestry, typed with the OncoArray or GWAS arrays, after high density imputation to the 1000 Genomes Project reference panel. They utilise a novel Bayesian fine-mapping algorithm that simultaneously dissects association signals within loci and constructs credible sets of variants driving the associations. Most loci contained a single association signal, although the maximum observed was five signals. Some loci were fine-mapped to relatively few variants. Multiple annotations were enriched for variants in the credible sets, including promoter and enhancer elements. Improved fine-mapping after annotation pointed to specific causal genes and mechanisms in some loci.

The study has been performed well, and implements a powerful and novel approach to fine-mapping. The manuscript is well written, and the methods and results are generally clearly described.

Major comments:

1. The approach a first selecting tags, fine-mapping tags, and then folding bag in variants that are in LD with the selected tags seems sub-optimal. I assume that this is done because it becomes difficult to distinguish between variants in strong LD and the MCMC algorithm gets “stuck” at one variant, rather than visiting the space of variants in very strong LD? Could the authors take the regions in which JAM indicated a single association signal, and compare the JAM credible set with that which would be obtained from using approximate Bayes’ factors (as described by Wakefield, AJHG 2007) with the meta-analysis results?

The reviewer is correct as to why we pruned variants in very high LD before running JAM. The reviewer also makes an interesting suggestion for checking the robustness of our results where a single signal is inferred. In regions with a single signal, credible sets from Wakefield’s approximate Bayes Factors, which ignores LD, should be nearly the same as those inferred by JAM, provided the degree of prior sparsity is equivalent. However, there is a difference in the prior set-up (we use a more sophisticated Beta-Binomial prior setup), so we could never exactly match the priors, and Wakefield’s method of course provides approximate Bayes Factors, whereas JAM provides formal Bayes Factors. Therefore, any differences in credible sets would be difficult to interpret without a detailed simulation study. This would certainly be interesting but we feel that a methodological comparison of this nature would be beyond the scope of this paper.

2. The authors claim that stepwise conditional analysis approaches are sub-optimal for identifying distinct association signals and subsequent fine-mapping. However, for the loci where JAM has concluded that there is a single association signal, would running a simple approximate conditional analysis in GCTA (using the same LD reference), conditioning on the lead SNP, also lead to the conclusion that there is a single causal variant (i.e. the residual association is not significant).

Whether GCTA includes a secondary variant where JAM has not would depend on the chosen P -value threshold for additional SNP inclusion, and how this compares to the significance levels used with JAM’s Bayesian model selection. There is no exact mapping between

Bayesian and frequentist significance thresholds, therefore it would be difficult to confirm whether any differences were due to a genuine additional signal. A detailed simulation exercise would be required to study and confidently interpret any differences between these approaches. We view this as a methodological comparison beyond the scope of this paper and, to our knowledge, such a comparison does not yet exist in the literature to refer to.

3. Are the results obtained from JAM consistent with other fine-mapping approaches that make use of summary data and LD references (such as FINEMAP), and what are the relative advantages of JAM?

Whilst the relative performance of contemporary fine-mapping approaches is important, a comparison between our preferred algorithm, JAM, and other methodologies has been conducted as part of the original JAM publication. We therefore believe that to attempt further comparison of different methods is beyond the scope of this study. Furthermore, the inclusion of this detail would detract from the primary and distinctive goal of this publication (the provision of rich fine-mapping data for PrCa to the scientific community to facilitate downstream research and clinical application), towards a greater focus on the comparison of fine-mapping programs, an aspect that is regularly embarked upon elsewhere. Summary data from the Onco-Array meta-analysis will however be available for the community to access on dbGAP, which would allow an interested researcher to re-analyse our data using any desired tools and conduct this comparison if desired.

It is also worth noting that in this study we are applying the fine-mapping algorithm to actual study data in which the number and identity of the true causal variants is not known *a priori*. A conclusive comparison of fine-mapping methods would require a detailed simulation study in which the “truth” is known, in order to confidently interpret the relative accuracy of conflicting results from different models. There would therefore not be any readily available, definitively informative metrics upon which to unequivocally judge the relative success of different algorithms using the same data; beyond simple evaluation of whether an algorithm had detected any variants previously presumed to be causal within a highly refined set of candidates. We have been able to achieve this measure with JAM by detecting within our credible set all of the few variants previously

regarded as likely to be causal (at *MSMB*, *RFX6* and *HOXB13*; as described within our results 'Multivariate fine-mapping from univariate summary statistics' and 'Fine-mapping Resolution' sections on pages 10 and 14), in addition to discernment of other highly plausible biological candidates in regions lacking previously identified strong candidates.

As regards the relative advantages of JAM versus other methodologies, we detail the reasoning behind our choice over alternative frameworks:

JAM over GCTA

It is well established that Bayesian search procedures perform much better than stepwise selection in high dimensional settings, therefore owing to the very large set of variants to analyse we did not use GCTA. We also point again to the original JAM paper in which extensive simulations demonstrated superiority in the rankings of variants compared to GCTA.

JAM over FINEMAP

Firstly, FINEMAP does not use a formal Bayesian search (it uses an approximate approach which, by design, excludes some variant combinations from the search). By contrast, JAM provides formal Bayes Factors from a full and formal Reversible Jump MCMC model search procedure. Secondly, FINEMAP assumes that at least one signal exists in each region (it places 0 prior weight on the null model). Due to the number of regions being fine-mapped, we wanted to allow for the possibility that some of these regions could be false positives; JAM can provide evidence for no signal.

JAM over Other Non-stochastic Bayesian approaches

Other Bayesian methods such as CAVIAR and CAVIARBF exhaustively assess all possible causal configurations. With dense genotype data, a prior assumption on the number of signals within each given region is necessary, otherwise computational requirements become prohibitive. We viewed such assumptions as too restrictive to apply to our dataset given the vastly improved power of our meta-GWAS (>144,000 individuals) in comparison to pre-existing knowledge derived from smaller studies. We therefore favoured JAM, since its stochastic search does not require *a priori* restrictions on the expected number of signals within any given region.

4. How sensitive are the results of JAM to the choice of samples for the LD reference? Do the results remain the same if a different random selection of cases/controls is used?

As we used a relatively large and unbiased sample panel for the LD reference (20,000 cases and 20,000 controls randomly selected from the OncoArray sub-cohort), results are not materially affected by the precise samples chosen. To investigate this matter, for 4 pilot regions we ran JAM 10 times using different panels of randomly selected cases and controls, finding >90% concordance of variants selected in the credible set across all 10 runs and variants with high posterior probabilities selected consistently throughout. We considered however that attempting to include details of these additional checks within the manuscript could lead to confusion as to the methodology we had used when generating the data we present, and therefore prefer to keep this information as an internal validation measure; although we would be prepared to include it at the discretion of the editor.

5. Little reference is made to the allele frequency distribution of credible set variants. Are most variants (or most of the posterior probability) ascribed to common variants? Does this give any additional insight into genetic architecture?

Only a small proportion of variants within the credible set are low frequency (of the 3,700 variants in the JAM credible set, only 48 have MAF <5% and 2 MAF <1%). This suggests that common variants primarily represent the likely causal variants at the majority of these GWAS loci. We do however still observe a number of specific instances in which the most likely candidate variants identified are low MAF (e.g. coding variants within the *HOXB13*, *FAM111A*, and *ANO7* genes). We have added additional description regarding the allele frequency distribution of the credible set to the 'Multivariate fine-mapping from univariate summary statistics' section of the results on page 10 and added an additional supplementary figure (Supplementary Figure 1).

Minor comments (specific line numbers provided):

331. Using the phrase "in a large European ancestry population" makes it sound like it is a single study in one population, rather than a meta-analysis – worth considering re-phrasing.

We have altered the phraseology to read “in a large European ancestry meta-analysis cohort” to prevent confusion.

336. Wasn't clear how the 53+25 regions correspond to the 86 distinct regions described in line 330.

To simplify, we have altered the phraseology of this sentence to read “At the time of the design of the OncoArray we specifically selected 46,500 SNPs for fine-mapping of known PrCa or multiple cancer (prostate and breast or ovarian cancer) risk regions.”.

342. Would be useful to have an additional sentence with some detail about the meta-analysis (rather than the reader having to look forward to the methods).

We have moved the brief summary of the composition of the meta-analysis that was previously in the following paragraph to this section of the results on page 8 for greater clarity.

376. Important to specify the LD reference used here (in addition to the methods).

We have added the LD reference panel used to this section of the results on page 9.

401. Presumably a posterior probability is generated by JAM for each variant – it wasn't entirely clear where this posterior was being used, rather than the presence/absence of a variant in the credible set.

JAM does generate posterior probabilities and Bayes Factors for each variant analysed as the reviewer notes, as well as for each combination of variants. The posterior probabilities provide useful information when attempting to prioritise variants within each region specific credible set for downstream investigations. We have formally integrated the SNP-specific posterior probabilities with the annotation features in a quantile regression framework. We provide the individual posterior probabilities, full annotations and quantile regression results of all variants selected in the credible set in Supplementary Table 2a, and describe the statistical results for three example regions within the “Fine-mapping Resolution” sub-section of our results to provide an example of how these attributes can be

used and interpreted. Furthermore, posterior probabilities of specific combinations of variants were used to pick the most representative set of tags when calculating the contribution to familial relative risk from multi-signal regions, as described in the ‘Estimating the contribution to FRR of PrCa from these GWAS loci’ section of the results and the ‘Proportion of familial risk explained’ section of the methods.

469. Not clear what is meant by “global level”?

This sentence relates to the overall results of the quantile regression analysis across all regions, rather than those within individual regions. In this section we are attempting to describe how the quantile regression analysis can help to highlight the specific functional annotations most regularly observed among variants with the highest statistical probability of association with PrCa. These selected annotations would therefore indicate potential shared functional mechanisms that underlie PrCa risk across a number of the susceptibility loci we have fine-mapped. We have re-phrased this to read “across all 75 regions” to make this meaning more clear.

484. Presumably, the enrichment procedure doesn’t take account of the posterior probability – is there any way this could be incorporated?

For the assessment of enrichment of annotations among our candidate variants by Fisher’s exact test, which the reviewers’ question relates to, the posterior probability of variants is indeed not taken into account and the analysis only considers overlap between annotation features and inclusion of variants in the credible set. As the reviewer notes however, the relative posterior probabilities of the variants carrying these annotations may also provide additional pertinent information beyond the simple inclusion of a variant in the credible set. This fact formed the rationale behind conducting our conditional quantile regression analysis, which considers all annotation features and variant posterior probabilities in the same model and indicates annotations enriched within the upper distributions of variant posterior probabilities. We therefore do make use of this information in the more advanced quantile regression analysis, but also provide the reader with the results from a more widely familiar Fisher’s test applied to each annotation individually, in an attempt to demonstrate the enrichment of specific annotations

that we observe clearly and accessibly to readers of multiple backgrounds. We also provide all variant posterior probabilities, quantile regression adjustments and full annotations in supplementary table 2a, to enable more detailed exploration of our results for specific variants by interested parties.

711. Trans-ethnic fine-mapping – how would the variable LD structure between ethnicities be taken account of in JAM?

Unfortunately, as JAM requires a reference LD dataset, it would not be possible to simultaneously analyse data from multiple ancestral populations using this methodology at the present time. Our suggestion in this section is for a prospective researcher with GWAS data from a non-European population to compare JAM results from their sample set with our European fine-mapping credible set, with the aim that the different LD patterns between populations may help further reduce the credible set at these less well refined regions through prioritisation of the intersected credible set from both populations.

855. If you compare the post QC set of variants with all variants with $MAF > 0.5\%$ in 1000 Genomes Europeans – what proportion have been “successfully imputed” and carried forward for fine-mapping analysis?

In Total 647,849 variants were imputed, approximately half of which were rare and excluded due to $MAF < 0.005$ (288,033 variants). Of the remaining 359,816 variants, a further 146,088 we excluded due to low QC indicators, generating the final set of 213,728 high confidence variants carried forward for the final analysis.

Reviewers' comments:

Reviewer #2 (Remarks to the Author):

The authors have clearly answered a number of my concerns and I'm quite happy about these areas now.

However, I asked the authors to provide results from their study on several specific areas that would be of interest to a wider audience than just those with an interest in Prostate Cancer. This more general insight into finemapping is important for a journal such as this and a missed opportunity given the size of their dataset compared to most GWAS. I am concerned they have misunderstood my intention. I hoped they would provide concrete results, but their response is often to include quite a vague comment in the Discussion section. So:

(i) I would like to see a clearer comparison between the results of this finemapping study with the previous one which would give some insight into the stability of such results. While they give some details in their "response to reviewers" they barely comment on this in the manuscript saying only that they consider the current study to be "the most detailed fine-mapping study to date for variant prioritisation"

But how different are the individual variants in each region now compared to previously, how many regions now include more signals and how many fewer? Have the number of explanatory signals in each region increased in some and decreased in others enough that the results from the current and previous study show little correlation in the number of independent signals in each region? Is it the regions with bigger signals, which might then be more easily mapped, that tend to be more robust from the last finemapping study to this? What I'm interested in getting at is how reliable the results of finemapping studies are. Presumably you can finemap stronger signals with a smaller sample size, but as the sample size increases do you detect ever smaller signals at each locus or do the higher penetrance loci simply stabilise?

(ii) How often in the multi-signal loci is the original tag SNP in strong LD with one 'finemapped' SNP and how often in weaker LD with several. Saying "within certain regions... whilst for others..." is so vague as to be useless. This needs quantification.

(iii) The authors note that "This indicates that regions with lower effect sizes and weaker evidence for association that require larger sample sizes for their detection are less likely to contain additional independent risk variants." But as I pointed out in my original review it could simply be that where the signal is weaker you have less power to detect multiple variants and so refine the original signal. This is, in part, why I'm interested in knowing how stable the results are as the sample size increases.

(iv) Reviewer 3 asked about how many of the variants in 1000 Genomes Europeans with $MAF > 0.5\%$ were imputed in this dataset. The authors didn't answer (they simply said how many variants they had, not how many 'common' ones they missed). I would like to see the answer to this.

Reviewer #3 (Remarks to the Author):

The authors have done a great job in responding to comments. I have one final comment regarding the coverage of variation in the fine-mapping regions after imputation.

In response to my minor comment, line 855 - the authors state that 213,728 high-quality variants (MAF>0.5%) were carried forward for analysis, out of ~360,000 imputed variants. This seems like quite a dramatic loss of variation, since most European low-frequency variants would be expected to be well imputed with the 1000 Genomes reference panel. Given that fine-mapping relies on having near complete variant coverage in a region, I think the authors should state somewhere in the text the proportion of variants across fine-mapping regions that are included in the analyses, and some discussion of the limitation of imputed data for fine-mapping.

Dear Referees,

We thank you for your helpful suggestions during review of our manuscript "Fine-mapping of Prostate Cancer Susceptibility Loci in a Large Meta-Analysis Identifies Candidate Causal Variants" (**NCOMMS-17-09702**).

We are pleased that the actions we have taken previously to address these queries were satisfactory, and appreciate these additional clarifications in instances where we had previously misinterpreted the level of detail requested. We have incorporated every suggestion within the manuscript text and through the inclusion of additional supplementary material, and have endeavoured to describe clearly our response to each comment individually below. We are confident that the additional material provided in relation to these suggestions would further improve the final publication for a broad audience.

We thank you for your time and look forward to your response.

Reviewers' comments:

Reviewer #2 (Remarks to the Author):

The authors have clearly answered a number of my concerns and I'm quite happy about these areas now.

However, I asked the authors to provide results from their study on several specific areas that would be of interest to a wider audience than just those with an interest in Prostate Cancer. This more general insight into finemapping is important for a journal such as this and a missed opportunity given the size of their dataset compared to most GWAS. I am concerned they have misunderstood my intention. I hoped they would provide concrete results, but their response is often to include quite a vague comment in the Discussion section. So:

(i) I would like to see a clearer comparison between the results of this finemapping study with the previous one which would give some insight into the stability of such results. While they

give some details in their "response to reviewers" they barely comment on this in the manuscript saying only that they consider the current study to be "the most detailed fine-mapping study to date for variant prioritisation"

But how different are the individual variants in each region now compared to previously, how many regions now include more signals and how many fewer? Have the number of explanatory signals in each region increased in some and decreased in others enough that the results from the current and previous study show little correlation in the number of independent signals in each region? Is it the regions with bigger signals, which might then be more easily mapped, that tend to be more robust from the last finemapping study to this? What I'm interested in getting at is how reliable the results of finemapping studies are. Presumably you can finemap stronger signals with a smaller sample size, but as the sample size increases do you detect ever smaller signals at each locus or do the higher penetrance loci simply stabilise?

We apologise for misinterpreting the level of detail desired to be presented in the previous query, and therefore our original response being more narrow in scope. Due to the number of regions we have interrogated in this study, we do not regard it as feasible to provide a region by region comparison of our latest results versus the iCOGS study within the text body of the manuscript. Instead we have endeavoured to make accessible a greater and more specific level of detail, through the inclusion of an additional supplementary table (Supplementary Table 6). This table would allow interested parties to compare the main results of the two studies at each individual region and discern general trends. We believe that all the information requested can be obtained from this new supplementary table. In addition we have substantially expanded the summary of the relative results from the two studies in our discussion, on pages 18-19, to attempt to describe the elements requested more explicitly.

Separately from the amendments we have made to our manuscript described above, we would also be happy to be contacted directly should you desire any further information that falls outside the main remit of this publication to help inform your own work. In addition, we would like to draw your attention to a prospective upcoming review article on fine-mapping by Professor Daniel J. Schaid that we understand will compare methodologies and findings from many

GWAS studies and therefore may explore many of the phenomena that form the basis of this query.

(ii) How often in the multi-signal loci is the original tag SNP in strong LD with one 'finemapped' SNP and how often in weaker LD with several. Saying "within certain regions... whilst for others..." is so vague as to be useless. This needs quantification.

As with the previous query, we apologise for originally interpreting this query as requesting only a concise synopsis of the general trend, or lack thereof, that we observed regarding the correlation between original GWAS index SNPs and independently associated signals identified through fine-mapping. Due to varying precise relationships across different regions, the trends do not readily partition into absolute groupings that are amenable to succinct summary. An additional complication is that some regions contain a single original GWAS hit, whilst others contained more than one. In general however, we have observed examples where the original tag variant is in LD with only one of the novel lead variants, and others where the original tag(s) are in partial to strong LD with variants representing multiple signals.

In order to incorporate the level of detail requested into the article, whilst also refraining from protracted and verbose description of each region individually within the text to retain clarity, we have now included an additional supplementary figure (Supplementary Figure 6). This figure depicts the correlation between the original index SNP(s) and tags selected in the credible set in heat-map form for every region containing multiple independent signals. We believe this provides the precise quantification requested in the most viable format. We have also modified this section of our discussion on page 18, to provide specific details of example regions for the different situations we observe.

(iii) The authors note that "This indicates that regions with lower effect sizes and weaker evidence for association that require larger sample sizes for their detection are less likely to contain additional independent risk variants." But as I pointed out in my original review it could simply be that where the signal is weaker you have less power to detect multiple variants and so refine the original signal. this is, in part, why I'm interested in

knowing how stable the results are as the sample size increases.

We thank you for the observation that we had omitted to state this alternative explanation in the text. We have modified the relevant section on page 19 of our discussion to add this counter viewpoint, alongside the additional detail regarding the stability of results between this study and the previous iCOGS fine-mapping project provided in response to comment (i).

(iv) Reviewer 3 asked about how many of the variants in 1000 Genomes Europeans with $MAF > 0.5\%$ were imputed in this dataset. The authors didn't answer (they simply said how many variants they had, not how many 'common' ones they missed). I would like to see the answer to this.

As this query overlaps with the single comment by Reviewer 3, we have responded to both suggestions together below for clarity and simplicity.

Reviewer #3:

The authors have done a great job in responding to comments. I have one final comment regarding the coverage of variation in the fine-mapping regions after imputation.

In response to my minor comment, line 855 - the authors state that 213,728 high-quality variants ($MAF > 0.5\%$) were carried forward for analysis, out of ~360,000 imputed variants. This seems like quite a dramatic loss of variation, since most European low-frequency variants would be expected to be well imputed with the 1000 Genomes reference panel. Given that fine-mapping relies on having near complete variant coverage in a region, I think the authors should state somewhere in the text the proportion of variants across fine-mapping regions that are included in the analyses, and some discussion of the limitation of imputed data for fine-mapping.

We apologise for misinterpreting the previous query as a question relating to what proportion of common variants in our dataset passed QC, rather than what proportion of common variants found in

1000 Genomes European (1KG EUR) samples were represented in our data as originally intended.

In regards to the question of how many common variants were excluded from our post-QC dataset, we have identified 227,793 common ($MAF \geq 0.05$) variants within the 1KG EUR dataset, of which 186,907 (82%) passed our QC procedures. Of the 40,886 common variants excluded during QC, 3056 were excluded during basic QC due to $INFO < 0.4$, whilst the vast majority, 37,830, were removed in the additional strict filtering step to exclude variants with greater genotype uncertainty (discordance of variant MAFs calculated based on “dosage” and “best-guess” genotypes $\geq 10\%$). Deeper investigation of these 37,830 variants revealed that 72% were situated within segments of the genome flagged as potentially lower accessibility to genotyping (either masked as low complexity by RepeatMarker, or excluded when applying the 1000 Genomes Phase 3 Strict Mask), whilst a further 12% had intermediate INFO score values (0.4-0.8).

We concur with the above statement that fine-mapping relies on high density variant coverage within a region; however exhaustive variant QC to remove low quality variants is also of paramount importance, in order to prevent potential false positive errors arising from the presence of imputation artefacts. Whilst there is no uniformly agreed upon QC procedure that is routinely applied to every fine-mapping study or dataset, we believe that the stringent criteria we have implemented here will increase the robustness of our results; as demonstrated by the high proportion of variants excluded during the additional internal consistency check that are located within repetitive or otherwise ambiguous segments of the genome, which are less amenable to accurate genotyping.

We had previously presented our QC procedures solely in the methods section without elaboration for conciseness. However, we agree with the suggestion above to provide more detailed statement of the proportion of variants retained during our analyses, and also that additional description of our methodology and appraisal of the potential benefits and limitations of this approach may be beneficial to the reader. We have therefore added the requested commentary to our discussion section on pages 21-22 and stated the number of variants excluded during QC and proportion of common European variants represented in our dataset to the methods section page 24.

REVIEWERS' COMMENTS:

Reviewer #2 (Remarks to the Author):

I'm grateful to the authors for taking the time to add the extra detail that I requested. I find it very interesting and am pleased to see that there is a fair degree of consistency between their current and previous finemapping studies. I'm quite happy with this now.

Reviewer #3 (Remarks to the Author):

I am happy that the authors have dealt with my comment and the issues raised by the other reviewer .